# Identify Then Recommend:
# Towards Unsupervised Group Recommendation

**Yue Liu**
Ant Group
National University of Singapore
`yueliu1990731@163.com`

**Shihao Zhu**
Ant Group
Hangzhou, China

**Tianyuan Yang**
Ant Group
Hangzhou, China

**Jian Ma**
Ant Group
Hangzhou, China

**Wenliang Zhong** [*]
Ant Group
Hangzhou, China

## Abstract

Group Recommendation (GR), which aims to recommend items to groups of users, has become a promising and practical direction for recommendation systems. This paper points out two issues of the state-of-the-art GR models. (1) The pre-defined and fixed number of user groups is inadequate for real-time industrial recommendation systems, where the group distribution can shift dynamically. (2) The training schema of existing GR methods is supervised, necessitating expensive user-group and group-item labels, leading to significant annotation costs. To this end, we present a novel unsupervised group recommendation framework named Identify Then Recommend (ITR), where it first identifies the user groups in an unsupervised manner even without the pre-defined number of groups, and then two pre-text tasks are designed to conduct self-supervised group recommendation. Concretely, at the group identification stage, we first estimate the adaptive density of each user point, where areas with higher densities are more likely to be recognized as group centers. Then, a heuristic merge-and-split strategy is designed to discover the user groups and decision boundaries. Subsequently, at the self-supervised learning stage, the pull-and-repulsion pre-text task is proposed to optimize the user-group distribution. Besides, the pseudo group recommendation pre-text task is designed to assist the recommendations. Extensive experiments demonstrate the superiority and effectiveness of ITR on both user recommendation (e.g., 22.22% NDCG@5 ↑) and group recommendation (e.g., 22.95% NDCG@5 ↑). Furthermore, we deploy ITR on the industrial recommender and achieve promising results. The codes are available on GitHub[2]. A collection (papers, codes, datasets) of deep group recommendation/intent learning methods is available on GitHub[3].

## 1 Introduction

Online platforms' expansion and information overload make finding pertinent content difficult. Recommendations help by supplying users with items that match their tastes and activities. This paper categorizes recommendations into User Recommendation (UR) and Group Recommendation (GR). UR, the conventional method, delivers personalized content for single users. Differently, GR targets user collectives, striving for a consensus to please most group members.

---

[*]Corresponding Author
[2]https://github.com/yueliu1999/ITR
[3]https://github.com/yueliu1999/Awesome-Deep-Group-Recommendation

38th Conference on Neural Information Processing Systems (NeurIPS 2024).

One essential technique of GR methods is the aggregation of the user groups. The traditional group recommendation methods are based on the heuristic rules [5, 6, 4], limiting the representation capability. To this end, DNN-based methods are proposed and achieve promising performance. Concretely, the learnable aggregate methods [7, 22, 27, 58, 76] are developed based on the attention mechanism [57]. Besides, (hyper-)graph neural networks [64, 32, 75, 23, 70] are built to model the side information and high-order collaborative information in the groups for comprehensive aggregation. Moreover, contrastive learning is utilized to encourage the alignment of different views and distill complementary information [46, 47, 61]. Although verified effective, the above methods rely on in advance given group information. Therefore, on another aspect, researchers [63, 68, 69] aim to develop group discovery capability of GR models by using group annotations.

However, this research highlights two critical problems of recent state-of-the-art methods. Firstly, the promising performance of these methods relies on a pre-defined and fixed number of user groups. Therefore, they can not handle group recommendations without giving the number of user groups, and unfortunately, the number of groups is usually unknown and dynamic in real-time industrial data. Secondly, the supervised training schema of existing GR methods requires extensive human annotations for user-group distribution and group-item interaction, easily leading to significant costs. Therefore, we raise a question that is important for research and practical scenarios. Can group recommendations be handled well without the annotations regarding user group information?

From this motivation, a novel unsupervised group recommendation framework termed Identify Then Recommend (ITR) is proposed by the group identification and the group self-supervised recommendation. Specifically, in the process of group identification, the area and density of each region are determined automatically. In contrast, regions with higher densities are more likely to be recognized as group centers. In comparison, those with lower densities are decision boundaries. Subsequently, a heuristic merge-and-split strategy is developed to consolidate similar user groups and to divide distinct user groups using the explore-and-exploit rule. In this manner, the user groups can be identified without the pre-defined number of groups and the group annotations. Next, at the self-supervised learning stage, we design two pre-text tasks to assist the group identification and recommendations. Besides, the pull-and-repulsion pre-text task is designed to optimize the user-group distribution for group identification. For the group recommendation, we construct the pseudo-group-item labels to guide the self-supervised learning of the GR model. In summary, through the two stages above, we empower the ITR model to handle group discovery and group recommendation even without group annotations, improving the applicability and performance of group recommendation. We conduct comprehensive experiments to demonstrate the superiority of ITR on both user recommendation and group recommendation. Moreover, ITR solves the practical problem in the industry scenario and is successfully deployed on the large-scale recommendation system, achieving promising results. The main contributions of this paper are summarized as follows.

- We propose a novel unsupervised group recommendation framework, ITR, which empowers the model to handle group discovery and group recommendation without annotations of user groups.

- The adaptive density estimation and heuristic merge-and-split strategies are designed to identify user groups automatically. Besides, the pull-and-repulsion pre-text task and pseudo-group recommendation pre-text task are presented to strengthen self-supervised learning.

- Extensive experiments demonstrate the proposed ITR's superiority and effectiveness. ITR has also been deployed on a large-scale industrial recommendation system, achieving promising results.

## 2 Related Work

The related work of this paper mainly contains two domains, including group recommendation and unsupervised clustering. Due to the page limitation of the main text, we provide a detailed introduction to related papers in Appendix 7.1.

## 3 Methodology

This section presents our proposed method. First, we provide the notations and task definitions. Second, we analyze and identify the limitations of existing group recommendations and group discovery. Last, we propose our solutions to address these challenges.

### 3.1 Basic Notation & Task Definition

In this paper, we denote $\mathcal{U} = \{u_1, u_2, ..., u_i, ..., u_n\}$, $\mathcal{T} = \{t_1, t_2, ..., t_i, ..., t_m\}$, and $\mathcal{G} = \{g_1, g_2, ..., g_i, ..., g_k\}$, as user set, item set, and group set, respectively. $n = |\mathcal{U}|$, $m = |\mathcal{T}|$, and $k = |\mathcal{G}|$ is the number of users, the number of items, and the number of groups, respectively. $\mathbf{A} \in \mathbb{R}^{n \times k}$ denotes the user-group distribution, where $\mathbf{a}_{ij} = 1$ represents that the user $u_i$ participates in the group $g_j$, and $\mathbf{a}_{ij} = 0$ represents the opposite case. Besides, $\mathbf{P} \in \mathbb{R}^{n \times m}$ is the user-item interaction, where $p_{ij} = 1$ denotes the user $u_i$ has the interaction with the item $t_j$, and $\mathbf{a}_{ij} = 0$ represents the opposite case. Moreover, $\mathbf{Q} \in \mathbb{R}^{g \times m}$ is the group-item interaction, where $q_{ij} = 1$ denote the group $g_i$ has the interaction with the item $t_j$, and $q_{ij} = 1$ denote the opposite case. The basic notations are summarized in Table 5 of Appendix.

For the conventional group recommendation, the methods aim to recommend items to the user groups. Mathematically, give the user-group distribution $\mathbf{A}$, and the training set of user-item interaction $\mathbf{P}$, group-item interaction $\mathbf{Q}$, the method aims to train a group recommendation model $\mathcal{F}$. After training, given the test groups, $\mathcal{F}$ can provide the potential item recommendation for the groups. Also, $\mathcal{F}$ usually has the ability of user recommendation and can provide candidate items for the test users.

### 3.2 Problem Analysis

By analyzing the above task definition, we notice that the existing group recommendation methods rely on the given user-group distribution $\mathbf{A}$. However, we argue that the user-group distribution $\mathbf{A}$ is hard to obtain in the real-world recommendation system. First, on the large-scale data of users, we can not know how many groups they will form. Second, it is hard to know which groups the user will join in, and this process should be dynamic. On another aspect, the previous methods are trained based on group-item interaction $\mathbf{Q}$. It is also a scarce resource. First, as mentioned above, the group information is hard to capture on large-scale data. Second, the interaction between groups and items is also dynamic and costs large human annotations. To this end, we raise an essential question for research and industry. Can group recommendations be handled well without the group annotations (i.e., user-group distribution $\mathbf{A}$ and group-item interaction $\mathbf{Q}$)?

This paper aims to deploy the proposed method in the real-time large-scale industrial recommendation system. On the open benchmarks, we admit that the annotations of users and groups have already existed and in the experiments, we remove them for the unsupervised experimental setting. We also admit that annotating these toy datasets may not be very expensive. However, the group assignments must change dynamically during training, especially at the early stage. However, note that on real-time large-scale data, the annotation costs a lot, and the distribution will shift dynamically. For example, in our scenario, the application contains 130 million page views and 50 million user views per day. The group assignment and annotation will be changed daily since it is a real-time recommendation. In addition, this is a newly launched application. Therefore, the activities of users will shift drastically, e.g., from new users to old users. We believe it will lead to large annotation costs and distribution shifts, and we aim to develop a pure, unsupervised group identification method for the user/group recommendation. In this scenario, performing several runs of clustering methods (the one that needs a pre-defined number of clusters) is not applicable since the search space will become very large, especially since we don't know the cluster number for the daily data. The current methods cannot deal with the data without a given number of clusters. For our proposed method, we just need one pass to determine the cluster number, which can fulfill the requirement of the daily data. The complexity of the multiple trials will be T times than our proposed method, where T denotes the time of trials. The details regarding the application can be found in Section 7.6.

### 3.3 Identify Then Recommend

From this motivation, we propose a novel unsupervised group recommendation framework termed Identify Then Recommend (ITR), which can automatically identify the user groups and conduct self-supervised group recommendation, even without giving the group number $k$ and the user-group distribution $\mathbf{A}$. It mainly consists of two modules, including the group identification module and the self-supervised group recommendation module as follows.

We briefly introduce the core idea and the primary designs of our proposed method before introducing the method part. Concretely, we aim to develop an unsupervised group recommendation method since we find the promising performance of existing state-of-the-art methods relies on the annotations

of the groups. The experimental evidence can be found in Figure 1. However, in the real-world scenario, the annotations regarding the group-item interactions and the user-group assignments are always not available. Labeling them in the real-time recommendation system is expensive and even impossible. To this end, we develop a pure, unsupervised group recommendation method, which can automatically discover the user groups and provide precise recommendations for them. Therefore, the core ideas of our methods are twofold, including the group identification and the group recommendation. For group identification, our initial solution is to adopt some existing clustering or community detection methods that do not require the number of clusters, e.g., DBSCAN, DeepDPM, etc. However, these methods can not automatically discover the user groups since they need other hyper-parameters, such as the radius and calculation of the density. Therefore, we design an adaptive density estimation method, which can automatically estimate the density of the samples. Then, for the group discovery, we propose a heuristic merge-and-split strategy to merge similar users into the group and split different user groups in the large group. Besides, the group embeddings are set as the learnable neural parameters and can be optimized during the learning process. Moreover, for the group recommendation part, the existing method can not deal with it in the unsupervised setting. We propose two pre-text tasks, including the pseudo group recommendation pre-text task and the pull-and-repulsion pre-text task. The pull-and-repulsion pre-text task aims to optimize the group embeddings by separating the different groups and pushing the samples together to the corresponding groups. Besides, for the pseudo group recommendation pre-text task, it generates the pseudo labels for the group recommendation task and guides the network to conduct group recommendation even without the precise annotations. By these designs, our proposed ITR is able to automatically discover the user groups and then provide precise group recommendations for them. Therefore, it can be applied in the real-time large-scale recommendation system.

### 3.3.1 Group Identification Module

In this section, we propose a group identification module (GIM) to discover the user groups on the user embeddings in an unsupervised manner. Given the user set $\mathcal{U} = \{u_1, u_2, ..., u_i, ..., u_n\}$, the item set $\mathcal{T} = \{t_1, t_2, ..., t_i, ...t_m\}$, and their interaction $\mathbf{P} \in \mathbb{R}^{n \times m}$, we first encode the user and item into the latent space and obtain the user embedding $\mathbf{U} \in \mathbb{R}^{n \times d}$ and the item embedding $\mathbf{I} \in \mathbb{R}^{m \times d}$, where $d$ denotes the dimensions of latent embeddings.

**Adaptive Density Estimation.** Then, we regard the user embeddings as the initial candidate centers of the user groups. We define the identified user groups and their embeddings as $\mathcal{G}' = \{g_1', g_2', ..., g_i', ..., g_{k'}'\}$ and $\mathbf{G}' \in \mathbb{R}^{k' \times d}$, respectively. Here, $k'$ denotes the number of user groups. Note that $k'$ is initialized as $n$ and will dynamically change. After initialization, the density of each user group is adaptively estimated. Concretely, take user group $g_i'$ as an example, we first calculate the minimal distance $d_{\min}^{(i)} = \min_j \text{dis}(i, j)$ and maximal distance $d_{\max}^{(i)} = \max_j \text{dis}(i, j)$ from it to all other groups, where $\text{dis}(i, j)$ denote the distance between the $i$-th group and the $j$-th group. Then, based on the quantile valuable $q$, we calculate the radius proposal as follows.

$$r_q = d_{\min} + (d_{\max}^{(i)} - d_{\min}^{(i)}) \times q, \tag{1}$$

where $r_q$ denotes the radius proposal with the quantile $q$. Subsequently, based on $r_q$, we estimate the density of the user group $g_i'$ as formulated.

$$\mu_i = \max_q \left( \left( \sum_j \mathbb{1}_{\text{dis}(i,j) \leq r_q} \right) / \pi r_q^2 \right), \tag{2}$$

where $\mathbb{1}_{\text{dis}(i,j)}$ is an indicator function, which equals 1 when $\text{dis}(i, j) \leq r_q$, and equals 0 when $\text{dis}(i, j) > r_q$. Besides, $\pi r_q^2$ denotes the area of the user group $g_i'$. In equation (2), the group with high density $\mu$, i.e., containing more users but with less area, is more likely to be recognized as the group center, and the opposite case is more likely to be recognized as the decision boundary.

**Heuristic Merge-and-split Strategy.** After obtaining the estimated density of each group candidate, this section proposes a heuristic merge-and-split strategy to merge similar user groups dynamically and split the distinct user groups. Specifically, we first sort the group candidates based on the density $\mu$ in descending order, and the group with high density will be processed first. This process can help our model skip some repetitive operations, therefore saving time. Then, we process each group in

order based on the idea of the explore-and-exploit rule. Concretely, we design a greedy parameter $\alpha$ to control the exploding rate. For each group, we conduct the exploring strategy with the probability of $\alpha \in (0, 1]$, while with the probability of $1 - \alpha$, we conduct the exploiting strategy. The greedy parameter $\alpha$ is calculated as follows.

$$\alpha = \exp\left(-s_{\text{explore}}^2/(s_{\text{all}} + 1)\right), \tag{3}$$

where $s_{\text{explore}}$ and $s_{\text{all}}$ denotes the explored step and the overall steps. By this setting, the exploring rate will decrease quickly when the explored steps increase, therefore encouraging the model to explore at the beginning steps and exploit after the beginning steps.

Next, we detail the exploring and exploiting strategies for our group identification task. For the exploring strategy, we aim to discover the new user group candidates based on the existing information. To achieve this target, we utilize the group embedding at this step $s$, denoted as $\mathbf{g}'^{(s)} \in \mathbb{R}^{1 \times d}$, and the group embedding at last step $s - 1$, denoted as $\mathbf{g}'^{(s-1)}$. Subsequently, the new user group is generated as follows.

$$\mathbf{g}'^{\text{new}} = \sigma \times \mathbf{g}'^{(s)} + (1 - \sigma) \times \mathbf{g}'^{(s-1)}, \tag{4}$$

where $\sigma \in \mathcal{N}(0.5, 0.5)$ is the balance parameter. $\mathcal{N}$ denotes the Gaussian distribution of the 0.5 mean and 0.5 standard deviation. The equation (4) implies that the closer to the group center at the current step (with the highest density at the current step) or to the group center at the last step (with the highest density at the last step), with the lower possibility that the area contains the new candidate group center. In this manner, we guide the model to explore the new possible group candidate $g'^{\text{new}}$. In addition, it will be added to the existing group candidate set: $\mathcal{G}' \leftarrow \{g_1', g_2', ..., g_i', ..., g_{k'}', g'^{\text{new}}\}$, $k' \leftarrow k' + 1$. Also, the adaptive density of $g'^{\text{new}}$ will be estimated through equation (1) and (2).

For the exploiting strategy, we aim to conduct a merge or split operation for the current user group. We take the group $g_i'$ as an example and denote its radius and corresponding density as $r_i$ and $\mu_i$, respectively. Recall that we have sorted the user groups, so $g_i$ is the unprocessed group center with the highest density. Therefore, we first conduct a merge operation for $g_i$ as follows.

$$\mathbf{g}_i' \leftarrow \frac{1}{\sum_j \mathbb{1}_{\text{dis}(i,j) \leq r_i}} \sum_k \mathbb{1}_{\text{dis}(i,k) \leq r_i} \mathbf{g}_k, \tag{5}$$

where $\mathbb{1}_{\text{dis}(i,j) \leq r_i}$ indicate all of the groups within the circle area with $r_i$ radius. In this manner, similar user groups are merged together into the group center with the highest density at the current step. However, we argue that the merged group may contain several isolated density peaks of the user groups. Therefore, in addition to the merge strategy, we conduct the split strategy as formulated.

$$\mathbf{g}_i' \leftarrow \frac{1}{\sum_j \mathbb{1}_{\text{dis}(i,j) \leq r_i}(1 - \mathbb{1}_{\text{dis}(i,j) > r_j})} \sum_k \underbrace{\mathbb{1}_{\text{dis}(i,k) \leq r_i}}_{\text{merge}} \underbrace{(1 - \mathbb{1}_{\text{dis}(i,j) > r_j})}_{\text{split}} \mathbf{g}_k, \tag{6}$$

where $\mathbb{1}_{\text{dis}(i,j) \leq r_i}$ represents that it merges the group candidates within the circle area with $r_i$ radius while $(1 - \mathbb{1}_{\text{dis}(i,j) > r_j})$ represents that it splits the group candidates whose circle area with $r_j$ radius can not contain the current user group $g_i$. By these designs, the user groups are first merged and then several isolated user groups are split. This merge-and-split process updates the existing group candidate set as follows.

$$\mathcal{G}' \leftarrow \{g_1', g_2', ..., \underbrace{\{g_i', ...\}}_{\text{merge-and-split}}, ..., g_{k'}'\}, k' \leftarrow k' - \sum_j \mathbb{1}_{\text{dis}(i,j) \leq r_i}(1 - \mathbb{1}_{\text{dis}(i,j) > r_j}) + 1. \tag{7}$$

In summary, in the proposed group identification module, we first estimate the adaptive density and the radius of each group candidate. Subsequently, the heuristic merge-and-split strategy dynamically identifies the user groups with the exploring and exploiting. It explores the new groups around the current decision boundary and exploits the current information to merge and split the group candidates. By these designs, our proposed method is empowered to identify groups without the scarce group annotation information. Next, we introduce how to utilize identified groups.

### 3.3.2 Self-supervised Group Recommendation Module

In this section, a self-supervised group recommendation module (SGRM) is proposed to utilize the discovered user groups and promote the group recommendation. It consists of two pre-text tasks, including the pull-and-repulsion task and the pseudo-group recommendation task.

**Pull-and-repulsion Pre-text Task.** After obtaining the discovered user groups $\mathcal{G}'$ and corresponding group center embeddings $\mathbf{G}' \in \mathbb{R}^{k' \times d}$, we aim to optimize the group center embeddings along with the user embeddings via the pull-and-repulsion pre-text task as formulated.

$$\min \mathcal{L}_{\text{PAR}} = \min \underbrace{\frac{1}{nk'} \sum_{i=1}^{n} \sum_{j=1}^{k'} \left\| \tilde{\mathbf{u}}_i - \tilde{\mathbf{g}}'_j \right\|_2^2}_{\text{pull term}} + \underbrace{\frac{-1}{(k'-1)k'} \sum_{i=1}^{k'} \sum_{j=1, j \neq i}^{k'} \left\| \tilde{\mathbf{g}}'_i - \tilde{\mathbf{g}}'_j \right\|_2^2}_{\text{repulsion term}}, \tag{8}$$

where $\tilde{\mathbf{u}}_i = \mathbf{u}_i / \|\mathbf{u}_i\|_2$ denotes the normalized user embeddings and $\tilde{\mathbf{g}}'_j = \mathbf{g}'_j / \|\mathbf{g}'_j\|_2$ denotes the normalized group embeddings. In equation (8), the pull term aims to pull the users to the user groups, while the repulsion term aims to push the distinct groups away. In this manner, the users and the identified groups are optimized in a unified framework during training, achieving better performance for both the user and group recommendations.

**Pseudo Group Recommendation Pre-text Task.** In addition to the pull-and-repulsion pre-text task, we develop a pseudo-group recommendation pre-text task to assist the group recommendation. Concretely, we first calculate the distance $\mathbf{D} \in \mathbb{R}^{n \times k'}$ between users and groups, and then estimate the user-group distribution $\mathbf{A}' \in \mathbb{R}^{n \times k'}$ as follows.

$$d_{ij} = \|\tilde{\mathbf{u}}_i - \tilde{\mathbf{g}}'_j\|_2, \ a'_{ij} = \mathbb{1}_{d_{ij} < (\sum \mathbf{D}/nk')}, \tag{9}$$

where $\tilde{\mathbf{u}}_i = \mathbf{u}_i / \|\mathbf{u}_i\|_2$ denotes the normalized user embeddings and $\tilde{\mathbf{g}}'_j = \mathbf{g}'_j / \|\mathbf{g}'_j\|_2$ denotes the normalized group embeddings. Subsequently, we further estimate the pseudo interactions $\mathbf{Q}' \in \mathbb{R}^{k' \times m}$ between the discovered user groups and the items as follows.

$$\mathbf{Q}' = (\mathbf{A}')^{\mathrm{T}} \mathbf{P}, \tag{10}$$

where $\mathbf{P} \in \mathbb{R}^{n \times m}$ denotes the user-item interaction and $\mathbf{A}' \in \mathbb{R}^{n \times k'}$ denotes the estimated user-group distribution. Next, we conduct a pseudo group recommendation pre-text task to enhance the performance of the model as follows.

$$\min \mathcal{L}_{\text{PGR}} = \min \frac{1}{k'm} \sum \left( \mathbf{G}' \mathbf{I}^{\mathrm{T}} - \mathbf{Q}' \right)^2, \tag{11}$$

where $\mathbf{I} \in \mathbb{R}^{m \times d}$ is the item embedding and $\mathbf{G}' \in \mathbb{R}^{k' \times d}$ denotes the discovered group embedding. In summary, in our proposed self-supervised group recommendation module, we conduct two pre-text tasks to refine the discovered user groups and further promote the group recommendation.

### 3.3.3 Overall Objective

We integrate the above two modules and provide the overall process of our proposed ITR model. The overall objective $\mathcal{L}_{\text{ITR}}$ of ITR consists three parts, including the loss of pull-and-repulsion pre-text task $\mathcal{L}_{\text{PAR}}$, the loss of pseudo group recommendation task $\mathcal{L}_{\text{PGR}}$, and the loss of user recommendation $\mathcal{L}_{\text{U2I}}$. $\mathcal{L}_{\text{U2I}}$ is a binary personalized ranking loss. $\mathcal{L}_{\text{ITR}}$ is formulated as follows.

$$\mathcal{L}_{\text{ITR}} = a \times \mathcal{L}_{\text{PAR}} + b \times \mathcal{L}_{\text{PGR}} + \mathcal{L}_{\text{U2I}}, \tag{12}$$

where $a$ and $b$ denote the trade-off parameters. The process of ITR is summarized in Algorithm 1.

BPR loss is a commonly used loss function in the recommendation. In our proposed method, we follow ConsRec for the BPR loss. And $\mathcal{L}_{\text{U2I}}$ is the same as the $\mathcal{L}_{\text{user}}$ in the ConsRec. It is formulated as $\mathcal{L}_{\text{user}} = -\sum_{u_s \in \mathcal{U}} \frac{1}{|\mathcal{D}_{u_s}|} \sum_{(j,j') \in \mathcal{D}_{u_s}} \ln \sigma(\hat{r}_{sj} - \hat{r}_{sj'})$, where $\mathcal{D}_{u_s}$ is the user-item training set sample for user $u_s$ and the $(j, j')$ denotes the user $u_s$ prefers observed item $i_j$ over unobserved item $i_{j'}$. For the sampling of positive and negative sample pairs, we also follow ConsRec, i.e., randomly

sampling from missing data as negative instances to pair with each positive instance. For the number of negative samples, ConsRec conducts experiments and analyses in Figure 8 of their original paper. For fairness, we keep the original setting of the ConsRec.

---

**Algorithm 1** Identify Then Recommend (ITR)

---

**Input**: user set $\mathcal{U}$; item set $\mathcal{V}$; user-item interaction $\mathbf{P}$; epoch number $E$; trade-off parameter $a, b$; range of quantile $q$.
**Output**: Trained ITR model.

1: **for** epoch $= 1, 2, ..., E$ **do**
2:     Encode user embeddings $\mathbf{U}$ and item embeddings $\mathbf{I}$.
3:     **if** At group identification stage **then**
4:         Initialize group center $\mathcal{G}'$ based on $\mathbf{U}$.
5:         Estimate the adaptive density via Eq. (1) and (2).
6:         **if** Exploring **then**
7:             Discover new user groups via Eq. (4).
8:         **else if** Exploiting **then**
9:             Conduct merge-and-split strategy via Eq. (6) and (7).
10:         **end if**
11:     **end if**
12:     Calculate $\mathcal{L}_{\text{PAR}}$ to conduct pull-and-repulsion task via Eq. (8).
13:     Calculate $\mathcal{L}_{\text{PGR}}$ to conduct pseudo group rec. task via Eq. (11).
14:     Calculate $\mathcal{L}_{\text{U2I}}$ to conduct user recommendation task.
15:     Minimize $\mathcal{L}_{\text{ITR}}$ to train the ITR model via Eq. (12).
16: **end for**
17: **Return** Well-trained ITR model.

---

### 3.3.4 Complexity Analyses

We conduct complexity analyses of our proposed ITR method. First of all, we define the number of users, the number of groups, and the average size of groups as $n$, $k'$, and $n/k'$, respectively. In the process of the adaptive density estimation, the time complexity and space complexity of calculating radius proposal for one group is $\mathcal{O}(1)$, $\mathcal{O}(k'^2)$, respectively, and for all groups is $\mathcal{O}(k')$, $\mathcal{O}(k'^2)$, respectively. Then, the time complexity and space complexity of density estimation for all groups are $k' \times n/k'$, $n \times k'$, respectively. Subsequently, at the heuristic merge-and-split strategy stage, for the explore step, it takes $\mathcal{O}(k')$ time complexity and $\mathcal{O}(1)$ space complexity, respectively. And for the exploit step, it takes $\mathcal{O}(k' \times n/k')$ time complexity, and $\mathcal{O}(nk')$ space complexity, respectively. Besides, for the proposed pseudo recommendation pre-text task, the time complexity and space complexity are $\mathcal{O}(n \times k')$, and $\mathcal{O}(n \times k')$, respectively. In addition, for the proposed pull-and-repulsion pre-text task, the time complexity and space complexity is $\mathcal{O}(nk' + k'^2)$, and $\mathcal{O}(nk')$, respectively. Moreover, for the BPR loss, the time complexity and space complexity are $\mathcal{O}(n)$ and $\mathcal{O}(n)$, respectively. Overall, the time complexity and space complexity of our proposed ITR method is $\mathcal{O}(k' + k' \times n/k' + k' + k' \times n/k' + n \times k' + nk' + k'^2 + n) \rightarrow \mathcal{O}(nk' + k'^2)$ and $\mathcal{O}(k'^2 + n \times k' + 1 + nk' + n \times k' + nk' + n) \rightarrow \mathcal{O}(nk' + k'^2)$, respectively. Therefore, our proposed method will not bring large memory and time costs since the complexity of our method is linear to the number of users.

## 4 Experiment

In this section, we aim to demonstrate the superiority of our proposed ITR model. First, we introduce the experimental setup, including the experimental environment, public benchmarks, and evaluation metrics, and compare baselines and implementation details. Second, we conduct experiments to support our motivation and demonstrate the problems in the existing group recommendation methods. Third, we show the superiority of the ITR model by comparing it with the recent state-of-the-art methods. Subsequently, we conduct ablation studies to demonstrate the effectiveness of the proposed modules in the ITR model. Moreover, we analyze the hyper-parameters and conduct A/B testing in the Appendix.

## 4.1 Experimental Setup

**Experimental Environment.** Experimental results are obtained from the server with four core Intel(R) Xeon(R) Platinum 8358 CPUs @ 2.60GHZ, one NVIDIA A100 GPU (40G), and the PyTorch platform. During training, we monitored the training process via the Weights & Biases.

**Public Benchmark.** We conduct experiments on two real-world public datasets, Mafengwo and CAMRa2011 [7]. Mafengwo is a tourism website where users can record their traveled venues and create or join group travel. CAMRa2011 is a real-world dataset containing individual users' and households' movie rating records.

**Evaluation Metric.** To evaluate the ITR model, we adopt two groups of metrics, including Hit Ratio@$x$ (HR@$x$) and Normalized Discounted Cumulative Gain@$x$ (NDCG@$x$), where $x \in \{5, 10\}$.

**Compared Baseline.** We compare ITR with twelve state-of-the-art GR methods including Pop [18], NCF [26], AGREE [7], MAML, GroupIM [53], HyperGroup [23], HCR [32], HHGR [75], MeLU, CubeRec [16], ConsRec [64], and CoHeat [30]. The introduction refers to the Section 7.1.

**Implementation Detail.** For the baseline methods, we use recorded results or adopt their original code to reproduce results. In our ITR model, we set $b$ as 10. And $a$ is set to 0.01 for Mafengwo and 10 for CAMRa2011, respectively. The range of $q$ is set as $\{0.1, 0.2, 0.3\}$. The learning rate is set as 0.001 for CAMRa2011 and 0.0001 for Mafengwo, respectively. All results are obtained from three runs.

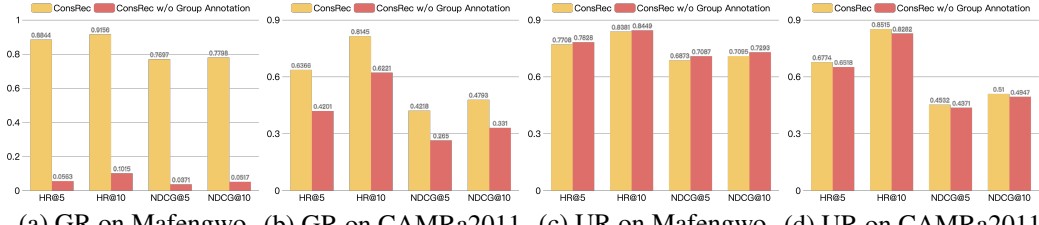

(a) GR on Mafengwo  (b) GR on CAMRa2011  (c) UR on Mafengwo  (d) UR on CAMRa2011

Figure 1: Motivation experiment on two datasets. "GR" and "UR" denote the group recommendation and the user recommendation, respectively.

## 4.2 Motivation Experiment

Recall the analyses in Section 3.2, we point out the issue of the existing group recommendation methods, i.e., the promising performance relies on the extensive group annotations, including the user-group distribution and the group-item interaction. To verify our claim and support our motivation, we conduct experiments on user recommendation tasks and group recommendation tasks on two datasets. Concretely, we test the performance of ConsRec [64] and it without the group annotations. The experimental results are presented in Figure 1, where the first and second rows denote the group recommendation and user recommendation tasks, respectively. From these results, we have the following conclusions. 1) For the group recommendation task, the performance decreases sharply when we remove the group annotations. For example, on the Mafengwo dataset, regarding to HR@5 metric, the performance drops from 0.8844 to 0.0563, leading to the failure of group recommendation. 2) The group annotations have little effect on the user recommendation task. In this multi-task learning process, the group annotations may have a positive effect (since they add information) or a negative effect (since they influence user recommendations we aim to develop a new method that can handle group recommendations even without group annotations.

## 4.3 Comparison Experiment

We conduct extensive comparison experiments with twelve baselines and our ITR model on two open datasets regarding two recommendation tasks. The experimental results are presented in Table 1 (for group recommendation) and Table 2 (for user recommendation). Based on these results, we find that 1) Firstly, regarding to group recommendation task, our proposed ITR model achieves the best results compared with the state-of-the-art baselines, even without giving the group annotations. For

Table 1: Group recommendation performance. **Bold** and underlined values denote the best and the runner-up. • denotes the model relies on group annotations. ° denotes the unsupervised model.

| Method | Mafengwo | | | | | CAMRa2011 | | | | |
|---|---|---|---|---|---|---|---|---|---|---|
| | HR@5 | HR@10 | NDCG@5 | NDCG@10 | Avg. | HR@5 | HR@10 | NDCG@5 | NDCG@10 | Avg. |
| Pop• | 0.3115 | 0.4251 | 0.2169 | 0.2537 | 0.3018 | 0.4324 | 0.5793 | 0.2825 | 0.3302 | 0.4061 |
| NCF• | 0.4701 | 0.6269 | 0.3657 | 0.4141 | 0.4692 | 0.5803 | 0.7693 | 0.3896 | 0.4448 | 0.5460 |
| AGREE• | 0.4729 | 0.6321 | 0.3694 | 0.4203 | 0.4737 | 0.5879 | 0.7789 | 0.3933 | 0.4530 | 0.5533 |
| MAML• | 0.4863 | 0.6484 | 0.3753 | 0.4383 | 0.4871 | 0.5974 | 0.7982 | 0.4123 | 0.4773 | 0.5713 |
| GroupIM• | 0.7377 | 0.8161 | 0.6078 | 0.6330 | 0.6987 | 0.6552 | **0.8407** | 0.4310 | 0.4914 | 0.6046 |
| HyperGroup• | 0.5739 | 0.6482 | 0.4777 | 0.5018 | 0.5504 | 0.5890 | 0.7986 | 0.3856 | 0.4538 | 0.5568 |
| HCR• | 0.7759 | 0.8503 | 0.6611 | 0.6852 | 0.7431 | 0.5883 | 0.7821 | 0.4044 | 0.4670 | 0.5605 |
| S2HHGR• | 0.7568 | 0.7779 | 0.7322 | 0.7391 | 0.7515 | 0.6062 | 0.7903 | 0.3853 | 0.4453 | 0.5568 |
| MeLU• | 0.5483 | 0.6626 | 0.4594 | 0.4726 | 0.5357 | 0.5763 | 0.7688 | 0.3827 | 0.4483 | 0.5440 |
| CubeRec• | 0.8613 | 0.9025 | 0.7574 | 0.7708 | 0.8230 | 0.6400 | 0.8207 | 0.4346 | 0.4935 | 0.5972 |
| ConsRec• | 0.8844 | 0.9156 | 0.7692 | 0.7794 | 0.8372 | 0.6407 | 0.8248 | 0.4358 | 0.4945 | 0.5990 |
| CoHeat• | 0.8992 | 0.9202 | 0.7924 | 0.8172 | 0.8573 | 0.6204 | 0.7974 | 0.4284 | 0.4820 | 0.5821 |
| ITR° | **0.9337** | **0.9377** | **0.8761** | **0.8775** | **0.9062** | **0.6952** | 0.7993 | **0.6653** | **0.6985** | **0.7146** |
| Impro. | 0.0345↑ | 0.0175↑ | 0.0837↑ | 0.0603↑ | 0.0490↑ | 0.0400↑ | 0.0414↓ | 0.2295↑ | 0.2040↑ | 0.1100↑ |

example, on the CAMRa2011 dataset, our ITR model achieves 0.2040 NDCG@10 improvement compared to the runner-up. The main reason is that the proposed GIM can precisely discover the potential user groups, and the proposed SGRM can take advantage of the group information to assist the group recommendation. 2) Secondly, for the user recommendation task, benefiting from the supplemental information of discovered user groups, our ITR also achieves significant improvement and the best performance. For example, on the Mafengwo dataset, we achieved 0.0474% NDCG@5 improvement compared to the runner-up. 3) Thirdly, on the CAMRa2011 dataset, for the HR@10 metric, our method can not beat the best methods. This could be due to the fact that GroupIM is unable to effectively capture the order of user preferences. While the items of interest to the user appear in the recommended list, these items are not ranked highly. In summary, this section verifies the superiority of our proposed ITR on both user recommendation and group recommendation. For the performance of our proposed method on the CAMRa2011 dataset, we consider it as a corner case since our proposed method can beat all the baselines with different metrics. And we want to give a reasonable explanation here. For the results, we can observe that our method can beat GroupIM with HR@5 but can not beat it with HR@10. And HR@5 is a more precise metric than HR@10 since it requires the model to rank correctly in the top 5 items. Therefore, we suspect that the ranking ability of GroupIM may not be strong and robust since it can achieve very promising performance when ranking in the top 10 items but can not beat our method when ranking in the top 5 items.

### 4.4 Ablation Study

In this section, we conduct ablation studies to verify the effectiveness of our proposed modules, including GIM and SGRM. The experimental results are shown in Table 3 (for group recommendation) and Table 4 (for user recommendation). Concretely, "Base", "Base+GIM+PAR", "Base+GIM+PGR", and "ITR" denote the baseline with group annotations, the baseline with GIM and pull-and-repulsive pre-text task, the baseline with GIM and pseudo group recommendation pre-text task, and our proposed method, respectively. From these results, we have the following observations. 1) "Base+GIM+PAR" achieves better performance compared with "Base", demonstrating the effectiveness of our proposed pull-and-repulsive pre-text task. 2) "Base+GIM+PGR" beats "Base", showing the effectiveness of our proposed pseudo group recommendation pre-text task. 3) Due to the GIM being merely a user group identification module, we can not use it to complete the recommendation task. But by combining observations 1) and 2), we find that GIM has contributed to the down-streaming tasks and brings performance improvement. 4) The combination variant "ITR" achieves the best performance.

## 5   Conclusion

This paper finds that the existing group recommendation methods rely on the pre-defined and fixed group number and the expensive user-group and group-item labels. To solve this problem and improve the applicability of group recommendation, we developed a novel unsupervised group recommendation method named ITR. It first identifies the user groups in an unsupervised manner

Table 2: Performance of user recommendation. **Bold** and underlined values denote the best and the runner-up. • denotes the model relies on group annotations. ° denotes the unsupervised model.

| | Mafengwo | | | | | CAMRa2011 | | | | |
|---|---|---|---|---|---|---|---|---|---|---|
| Method | HR@5 | HR@10 | NDCG@5 | NDCG@10 | Avg. | HR@5 | HR@10 | NDCG@5 | NDCG@10 | Avg. |
| Pop• | 0.4047 | 0.4971 | 0.2876 | 0.3172 | 0.3767 | 0.4624 | 0.6026 | 0.3104 | 0.3560 | 0.4329 |
| NCF• | 0.6363 | 0.7417 | 0.5432 | 0.5733 | 0.6236 | 0.6119 | 0.7894 | 0.4018 | 0.4535 | 0.5642 |
| AGREE• | 0.6357 | 0.7403 | 0.5481 | 0.5738 | 0.6245 | 0.6196 | 0.7897 | 0.4098 | 0.4627 | 0.5705 |
| MAML• | 0.4528 | 0.5187 | 0.3286 | 0.3579 | 0.4145 | 0.5102 | 0.6309 | 0.3523 | 0.4286 | 0.4805 |
| GroupIM• | 0.1608 | 0.2497 | 0.1134 | 0.1420 | 0.1665 | 0.6113 | 0.7771 | 0.4064 | 0.4606 | 0.5639 |
| HyperGroup• | 0.7235 | 0.7759 | 0.6722 | 0.6894 | 0.7153 | 0.5728 | 0.7601 | 0.4410 | 0.5016 | 0.5689 |
| HCR• | 0.7571 | 0.8290 | 0.6703 | 0.6937 | 0.7375 | 0.6262 | 0.7924 | 0.4195 | 0.4734 | 0.5779 |
| S2HHGR• | 0.6380 | 0.7520 | 0.4637 | 0.5006 | 0.5886 | 0.6153 | 0.8173 | 0.3978 | 0.4641 | 0.5736 |
| MeLU• | 0.7694 | 0.8358 | 0.7095 | 0.7256 | 0.7601 | 0.6092 | 0.7924 | 0.4376 | 0.4874 | 0.5817 |
| CubeRec• | 0.1847 | 0.3734 | 0.1099 | 0.1708 | 0.2097 | 0.5754 | 0.7827 | 0.3751 | 0.4428 | 0.5440 |
| ConsRec• | 0.7725 | 0.8404 | 0.6884 | 0.7107 | 0.7530 | 0.6774 | **0.8412** | 0.4568 | 0.5104 | 0.6215 |
| CoHeat• | 0.7023 | 0.7621 | 0.6035 | 0.6492 | 0.6793 | 0.6282 | 0.7542 | 0.4015 | 0.4495 | 0.5584 |
| ITR° | **0.8107** | **0.8586** | **0.7569** | **0.7727** | **0.7997** | **0.7106** | 0.8017 | **0.6790** | **0.7083** | **0.7249** |
| Impro. | 0.0382↑ | 0.0182↑ | 0.0474↑ | 0.0471↑ | 0.0397↑ | 0.0332↑ | 0.0395↓ | 0.2222↑ | 0.1979↑ | 0.1034↑ |

Table 3: Ablation studies of GIM and SGRM on group recommendation task.

| | Mafengwo | | | | | CAMRa2011 | | | | |
|---|---|---|---|---|---|---|---|---|---|---|
| Method | HR@5 | HR@10 | NDCG@5 | NDCG@10 | Avg. | HR@5 | HR@10 | NDCG@5 | NDCG@10 | Avg. |
| Base | 0.8844 | 0.9156 | 0.7692 | 0.7794 | 0.8372 | 0.6407 | 0.8248 | 0.4358 | 0.4945 | 0.5990 |
| Base+GIM+PAR | 0.9065 | 0.9156 | 0.8152 | 0.8183 | 0.8639 | 0.6807 | 0.8165 | 0.6345 | 0.6786 | 0.7026 |
| Base+GIM+PGR | 0.8864 | 0.8975 | 0.7934 | 0.7971 | 0.8436 | 0.6497 | 0.8097 | 0.5559 | 0.6082 | 0.6559 |
| ITR | 0.9337 | 0.9377 | 0.8761 | 0.8775 | 0.9062 | 0.6952 | 0.7993 | 0.6653 | 0.6985 | 0.7146 |

Table 4: Ablation studies of GIM and SGRM on user recommendation task.

| | Mafengwo | | | | | CAMRa2011 | | | | |
|---|---|---|---|---|---|---|---|---|---|---|
| Method | HR@5 | HR@10 | NDCG@5 | NDCG@10 | Avg. | HR@5 | HR@10 | NDCG@5 | NDCG@10 | Avg. |
| Base | 0.7725 | 0.8404 | 0.6884 | 0.7107 | 0.7503 | 0.6774 | 0.8412 | 0.4568 | 0.5104 | 0.6215 |
| Base+GIM+PAR | 0.7788 | 0.8347 | 0.7125 | 0.7310 | 0.7643 | 0.6970 | 0.8316 | 0.6431 | 0.6866 | 0.7146 |
| Base+GIM+PGR | 0.7959 | 0.8432 | 0.7235 | 0.7392 | 0.7755 | 0.6811 | 0.8253 | 0.5781 | 0.6249 | 0.6774 |
| ITR | 0.8107 | 0.8586 | 0.7569 | 0.7727 | 0.7997 | 0.7106 | 0.8017 | 0.6790 | 0.7083 | 0.7249 |

and then conducts two pre-text tasks to promote the recommendation. For the group identification, we calculate the adaptive density of groups to determine the group centers and decision boundaries aa and then conduct a heuristic merge-and-split strategy to discover new groups and merge similar user groups. For the self-supervised group recommendation, we design the pull-and-repulsion pre-text task to optimize the users and groups in a unified framework. Besides, a pseudo group recommendation task is designed by estimating user-group distribution and group-item interactions to assist user recommendation and group recommendation. Extensive experiments demonstrate the superiority and effectiveness of our proposed method on both user recommendation tasks and group recommendation tasks. Benefiting the superiority of ITR, it is also deployed on the large-scale industrial recommendation system and achieves promising improvements. However, ITR still relies on the user-item interaction for the user recommendation and group recommendation. In the future, we aim to develop an unsupervised recommendation method or the zero-shot recommendation method to solve the data sparsity problem.

# 6 Acknowledgment

We thank all anonymous reviewers for their constructive and helpful reviews. This work was supported by the National Natural Science Foundation of China (No. 62325604 and 62276271).

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

# 7 Appendix

## 7.1 Related Work

### 7.1.1 Group Recommendation

Group Recommendation (GR), which aims to recommend items to user groups, is a practical yet challenging task [56]. Typically, methods learn user embeddings and then aggregate the information into groups [73, 54, 28]. For the aggregation, traditional methods are based on heuristic rules, e.g., average [5], least misery [6], and maximum satisfaction [4].

Recently, deep learning methods [29, 49] have shown promising performance. Attention-based aggregation techniques [15, 60, 31] learn agreement weights through attention mechanisms. At first, AGREE [7] adopts the neural attention network to adapt the group embedding. Subsequently, [8, 77, 22] enhance attention by incorporating social information. MoSAN [58] represents a single user with a sub-attention module, which learns the user's preferences from all other group members. Besides, GAME [27] is introduced to learn independent and counterpart views based on the interaction graph. [76] adopt the dependency relationship between items to enhance the interactions.

Subsequently, graph-neural-network-based methods [20, 13, 14, 37, 71, 78] capture the side and higher-order information in groups for better aggregation. For example, [32] developed a dual

channel hyper-graph convolutional network to capture the member-level and group-level preferences. Similarly, a hierarchical hyper-edge embedding-based group recommender named Hypergroup is proposed by [23]. In addition, [75] proposes HHGR based on a hierarchical hyper-graph convolutional network and double-scale node dropout strategy. Furthermore, [70] proposes MMAN by designing the GNN-based encoder and neural-based aggregating networks. ConsRec [64] builds three distinct views that provide mutually complementary information to enable multi-view learning.

Moreover, self-supervised learning [79, 66, 62, 72] becomes a valuable training paradigm to improve the representation learning capability of group recommendation methods via pre-text tasks [33, 36]. Concretely, crossCBR [46] designs the cross-view contrastive learning method by encouraging the alignment of the two separately learned views. GroupIM [53] aims to overcome data sparsity by maximizing mutual information between groups and group members. Besides, [47] utilizes a strategy that first fuses the multi-view representations before performing self-supervised learning. In addition, a dual-supervised contrastive learning method named DSCBR [61] is proposed to integrate supervised learning and self-supervised learning. Furthermore, [16] replaces the point representations with hyper-cubes to improve flexibility and capacity to account for the multi-faceted user preferences. Moreover, [30, 28, 38, 3] are proposed to solve the cold-start problem in the group recommendation. [74] solve group buying problems via multi-task learning.

Although verified effective, the above methods rely on given group information in advance. Therefore, researchers aim to empower GR models with the group discovery capability by utilizing existing group annotations. Specifically, [68] CFAG is proposed to conduct group identification based on the tripartite graph convolution layers and the propagation augmentation layer. In addition, DiRec [63] is proposed by exploiting the social intent and interest-intent of users, better motivating them to join groups. Moreover, in GTGS [69], researchers design transitional hyper-graph convolution layer and cross-view self-supervised learning to improve the performance.

However, two problems with existing GR methods limit their performance and applicability. (1) The number of user groups is predefined and fixed. (2) The training scheme requires expensive group annotations, including the user-group distribution and the group-item interaction. Therefore, we develop an unsupervised group recommendation framework ITR to solve them.

### 7.1.2 Unsupervised Clustering

Unsupervised clustering [? ] is a fundamental and challenging task that aims to group samples into distinct clusters without supervision. By exploiting the power of unlabelled data, clustering algorithms have found applications in various domains, including computer vision [12], natural language processing [2], graph learning [44], recommendation systems [45], and community detection [55]. In the early stages, several traditional clustering methods were proposed. For example, classical $k$-means clustering [25] iteratively updates cluster centers and assignments to group samples. Spectral clustering [59] constructs a similarity graph and uses eigenvalues and eigenvectors to perform clustering. In addition, GMM [50] assumes that the data distribution is a mixture of Gaussians and estimates parameters by maximum likelihood. In addition, repulsive clustering methods [34, 19, 1] cluster the data using the repulsive terms. In contrast, density-based methods [21, 51, 17] overcome the need to specify the number of clusters as a hyperparameter. In recent years, the impressive performance of deep learning has led to a growing interest in deep clustering [52, 48, 35]. For example, Xie et al. propose DEC [65], a deep learning-based approach to clustering. They initialize cluster centers using $k$-means clustering and optimize the clustering distribution using a Kullback-Leibler divergence clustering loss [65, 24]. JULE [67] and DeepCluster [9] both take an iterative approach, updating the deep network based on learned data embeddings and clustering assignments. SwAV [10], an online method, focuses on clustering data and maintaining consistency between cluster assignments from different views of the same image. DINO [11] introduces a momentum encoder to address representation collapse. Moreover, DCRN and IDCRN models introduced a dual correlation reduction strategy to tackle the issue of representation collapse [40, 41]. Additionally, the HSAN framework employed a dynamic weighting strategy to effectively mine challenging sample pairs [? ]. The SCGC model focused on simplifying graph augmentation by utilizing parameter-unshared Siamese encoders along with embedding disturbance [? ]. Meanwhile, the TGC framework presented a versatile approach for deep node clustering specifically designed for temporal graphs [? ]. Despite their contributions, many of these methods struggle to handle large graphs containing millions of nodes. To address this scalability issue, approaches like $S^3GC$ [? ] and Dink-Net [43] were developed. Furthermore, the RGC algorithm was introduced to address the challenge

of unknown cluster numbers through reinforcement learning [42]. In our research, we propose to develop an unsupervised group recommendation framework that clusters users into distinct groups and subsequently delivers personalized recommendations tailored to these user groups.

## 7.2 Notation

We list the basic notations in Table 5.

Table 5: Basic notations.

| Notation | Meaning | Notation | Meaning |
|---|---|---|---|
| $\mathcal{U}$ | user set | $n$ | number of users |
| $\mathcal{T}$ | item set | $m$ | number of items |
| $\mathcal{G}$ | group set | $k$ | number of groups |
| $\mathcal{G}'$ | identified group set | $d$ | dimension of latent features |
| $\mathbf{A} \in \mathbb{R}^{n \times k}$ | user-group distribution | $q$ | quantile valuable |
| $\mathbf{A}' \in \mathbb{R}^{n \times k'}$ | estimated user-group distribution | $r_q$ | radius proposal with quantile $q$ |
| $\mathbf{P} \in \mathbb{R}^{n \times m}$ | user-item interaction | $\mu_i$ | estimated density of $g'_i$ group |
| $\mathbf{Q} \in \mathbb{R}^{k \times m}$ | group-item interaction | $\alpha$ | greedy parameter |
| $\mathbf{Q}' \in \mathbb{R}^{k' \times m}$ | estimated group-item interaction | $\mathcal{L}_{\text{PAR}}$ | pull-and-repulsion loss |
| $\mathbf{U} \in \mathbb{R}^{n \times d}$ | user embedding | $\mathcal{L}_{\text{PGR}}$ | pesudo group recommendation loss |
| $\mathbf{I} \in \mathbb{R}^{m \times d}$ | item embedding | $\mathcal{L}_{\text{U2I}}$ | user recommendation loss |
| $\mathbf{G} \in \mathbb{R}^{k \times d}$ | group embedding | $\mathcal{L}_{\text{ITR}}$ | overall loss of ITR method |
| $\mathbf{G}' \in \mathbb{R}^{k' \times d}$ | identified group embedding | | |

## 7.3 Hyper-parameter Analysis

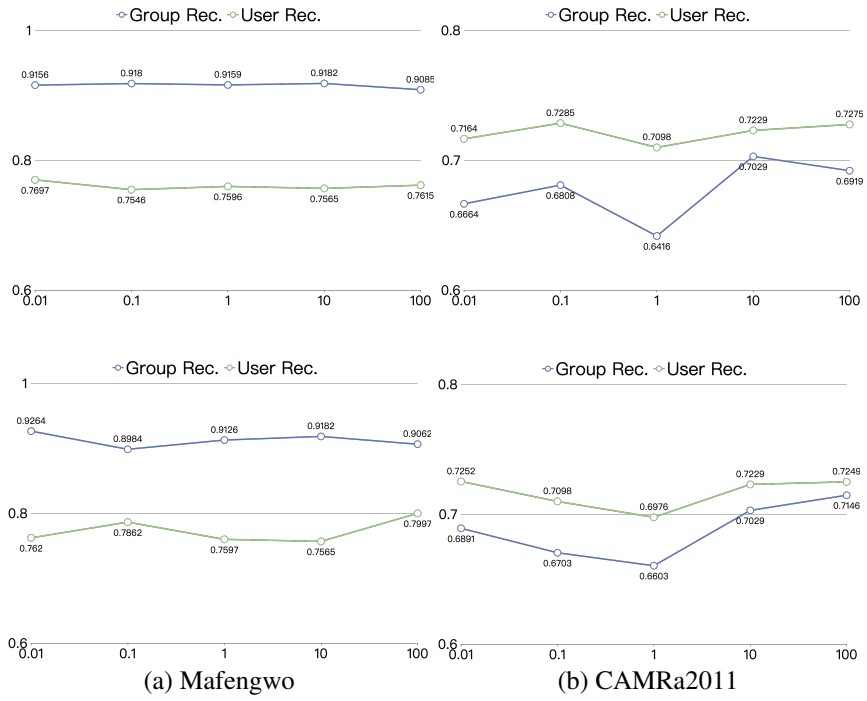

(a) Mafengwo        (b) CAMRa2011

Figure 2: Hyper-parameter analysis. The first and second row denotes trade-offs $a$ and $b$, respectively.

We analyze the hyper-parameters in ITR in this section. Concretely, we first fix $a = 10$ and then tune $b \in \{0.01, 0.1, 1, 10, 100\}$. Besides, we fix $b = 10$ and then tune $a \in \{0.01, 0.1, 1, 10, 100\}$. The experimental results are shown in Figure 2, where the first and second row denotes the results

of trade-off $a$ and $b$, respectively. From these results, we have the following conclusions. 1) Our proposed ITR is not very sensitive to trade-off $a$ on the Mafengwo dataset. And achieve promising performance on the CAMRa2011 dataset when $a \in [10, 100]$. 2) For the trade-off $b$, the results shown that ITR can perform well when $b \in [10, 100]$.

## 7.4 Effectiveness of GIM

The group identification module (GIM) aims to discover the user groups based on the embeddings in the pure, unsupervised setting. In the pure unsupervised setting, the GIM is a must-used module to discover the user groups. And the effectiveness can be verified indirectly by the performance improvement of downstream tasks. To directly demonstrate the effectiveness of the proposed group identification module, we follow your suggestion and verify the quality of the discovered user group. Concretely, on the open benchmarks, we adopt the unsupervised clustering metric Silhouette Coefficient (SC) to evaluate the quality of the discovered user groups. SC is an important indicator for evaluating clustering quality, which can help evaluate the rationality of clustering results. It combines the cohesion and separation of each data point to quantify the clustering effect, and its value range is from -1 to 1. Specifically, the larger the SC, the better the clustering effect of the data points. On the average of the four datasets used in our paper, the discovered user groups achieve 0.879 SC, demonstrating the promising clustering performance of our proposed GIM. On the industrial data, due to the labels are not available and the data size being large, we conducted case studies to demonstrate the effectiveness of GIM. Specifically, we select some hot livestream rooms for the case studies, e.g., the gold-selling livestream room, movie-ticket-selling livestream room, and the face-tissue-selling livestream room. Then, we check the user group distribution and find the major group. Subsequently, we use some tags of the user interests to provide the labels of the major groups. In this manner, we can determine the interest of the major groups in the studies livestream room. We discuss the results as follows. For the gold-selling livestream room, the interests of the major groups include jewelry & gold and financial management. Besides, for the movie-ticket-selling lives interests of the major groups include movies and entertainment. Differently, for the face-tissue-selling livestream room, the interests of the major groups include food and hygiene products. These case studies can demonstrate that our proposed GIM method can effectively group the similar users to one group and separate the different groups. Also, the captured information of the major group can also be used for the refined recommendation.

## 7.5 Convergence

For convergence, we checked the loss and performance of our method on all datasets and found that they converge well, i.e., the loss decreases and gradually converges to a low value, while the performance increases and gradually converges to a high value.

## 7.6 Application

### 7.6.1 Application Background

We apply the propose ITR method in the livestreaming recommendation scenario. In a liveslive streaming, the users naturally form several user groups based on their interests. For instance, in a sports goods life streaming, the users with basketball interests will form a user group, while the users with badminton interests will form another user group. These user groups are important information to describe the profile of the live streaming. For example, when this live streaming begins to sell basketball-related goods, the basketball user group will be activated and be willing to click, comment, or buy merchandise. Therefore, we should guide more users with basketball interests into this live streaming. Following this principle, we aim to group the users in the live streaming live streaming different groups and utilize this group information to assist the recommendation. However, the number of user groups is always unknown and dynamic. Besides, the different live streaming rooms have different group numbers, e.g., the gold live streaming room has fewer group numbers than the groceries live streaming room since user interests in the gold live streaming room are more concentrated. In addition, the group annotation information, such as user-group distribution and group-item interactions, is also missing or needs extensive human annotation costs in this scenario. To solve these problems, we present the ITR model in this paper. The proposed GIM can identify the user groups in the live streamings, and the proposed SGRM can assist and promote the recommendation.

Furthermore, upon acquiring the identified group embeddings, we can also utilize them to enhance the traffic mechanism within real-time livestreaming recommendations. In practical terms, we initially designate the primary user groups within live streaming as the room's profile. Additionally, we devise and calculate the matching score between the primary user groups and potential users, subsequently leveraging these matching scores to dynamically bolster the ranking score of relevant users, therefore improving the recommendation performance and the interactive streaming live streaming room.

For the click metrics, there are for MCS, including uv (user view), uvctr (user view click-through), PV (page view), and pvctrand (page view click-through). And for the trade metrics, there are two metrics, including iv (user view) and pPV(page view). For the data scale, the application contains about 130 million page views and 50 million users per day. And the baseline model is the MMOE model for recommendation. We merely compared the baseline model with our strategy and the baseline model without our strategy to demonstrate the effectiveness of our proposed method.

The baseline model is the conventional MMOE-based recommendation model. We aim to adopt our proposed method at the re-rank stage, i.e., modify the predicted score of the recommendation model by considering the user groups. Next, we assume you have carefully read the background of this application and introduce the details of adopting our method into the re-rank stage. For different live streamings, the number of user groups will be different. For example, the gold live streaming room has fewer group numbers than the groceries live streaming room since user interests in the gold live streaming room are more concentrated. Therefore, we need an unsupervised group recommendation method to discover the user groups in different live streamings. Here, we adopt our ITR method to discover the user groups and then produce the group embeddings. Then upon acquiring the identified group embeddings, we can also utilize them to enhance the traffic mechanism within real-time livestreaming recommendations. In practical terms, we initially designate the primary user groups within live streaming as the room's profile. Additionally, we devise and calculate the matching score between the primary user groups and potential users, subsequently leveraging these matching scores to dynamically bolster the ranking score of relevant users, therefore improving the recommendation performance and the interactive environments in the live streaming room. And oOurosed ITR will not be used with the group annotations since if the group annotations are easy to obtain, we don't need to develop an unsupervised learning-based group recommendation method.

### 7.6.2 A/B Testing

To further demonstrate the effectiveness of ITR, A/B testing is conducted in this section. The experimental results shown in Table 6 indicate that ITR can improve the baseline model by about $2.45\%$ performance on average, showing the effectiveness of our ITR model.

Table 6: A/B testing on real-time industrial recommender. The symbol "-" denotes business secret.

| Method | Click Metrics | | | | Trade Metrics | |
|---|---|---|---|---|---|---|
| | uv | uvctr | pv | pvctr | uv | pv |
| Base | - | - | - | - | | |
| Base+ITR | 2.78% ↑ | 2.37% ↑ | 2.75% ↑ | 2.09% ↑ | 2.96% ↑ | 1.76% ↑ |

### 7.7 Alignment between Contribution and Results

In our view, we think unsupervised learning is a setting that is more challenging compared with supervised learning. Correct me here if you have different opinions. Therefore, the unsupervised user/group recommendation is more challenge compared with the supervised ones (see the experimental results in Figure 1). In this paper, we aim to point out that most of the existing deep group recommendation methods are supervised, but in real-world scenarios, the labels are always missing. Therefore, we need to process the user/group recommendation in an unsupervised manner. In addition, with the unsupervised learning setting, we develop a novel group recommendation method by identify-then-recommend schema. Concretely, the designed group identification module aims to discover the user groups on the user embeddings. The proposed pseudo-group recommendation pre-text task and the pull-and-repulsion pre-text task aim to produce high-quality group embeddings and conduct precise recommendations, respectively. Besides, thanks for your suggestion on the experiment. However, the variant of "known group annotation + GIM + PAR + PGR" may not be reasonable since if we already know the group annotation, we don't need to conduct the group

identification. Therefore, if the group annotation is missing (the settings of this paper), we must use the GIM and then conduct the ablation studies on the PAR and PGR. The GIM is a must-use module, and the effectiveness of GIM can be verified through the performance of the downstream tasks. The experimental results can be found in Table 3 and Table 4.

## 7.8 Advantage of Our Clustering Method

The clustering methods, which need a pre-defined number of cluster centers, such as k-means and spectral clustering, cannot be applied in this scenario. For other clustering methods, which do not require the pre-defined number of cluster centers, such as DBSCAN, it is hard to deal with our scenario well since 1) the data distribution of changes plication changes dynamically; therefore, it's hard to determine the parameters regarding the density, such the radius, and the threshold. But our proposed group identification module can automatically estimate the density and discover the groups via the heuristic merge-and-split strategy, thus can be easily applied in the real-time large-scale unsupervised recommendation system. 2) They can not be integrated into our framework without our proposed pre-text tasks, including the pseudo group recommendation pre-text task and the pull-and-repulsion pre-text task.

In this paper, the setting of the user/group recommendation is pure unsupervised and the cluster number is not pre-given in both the open dataset and the industrial data. Therefore, during the training process, the cluster number will dynamically change on the open benchmarks since the status of the user embeddings changes. Our proposed GIM can automatically discover and optimize the group embedding and the group number. After training, the number of clusters and the status of the group embeddings are restored, and then they will be used to conduct the testing of the group recommendation and the user recommendation.

Besides, on the real-time large-scale data, we first train the recommendation model using the past data. During this process, the number of clusters and the status of the group embeddings are optimized dynamically. After the training stage, they are restored to the database for serving the downstream tasks. Then, when the training data is updated, the recommendation model will be continued trained, and the number of clusters also change dynamically.

## 7.9 Difference from Occasional Group Recommendation Problem

In our paper, we claimed that two critical problems limit the applicability of the recent state-of-the-art methods are twofold. Firstly, the promising performance of these methods relies on a pre-defined and fixed number of user groups. Therefore, they can not handle group recommendations without giving the number of user groups, and unfortunately, the number of groups is usually unknown and dynamic in real-time industrial data. Secondly, the supervised training schema of existing GR methods requires extensive human annotations for user-group distribution and group-item interaction, easily leading to significant costs. The previous methods either rely on the pre-defined cluster number or the user-group / group-item annotations. Note that the setting of our experiments is purely unsupervised, and we are not merely solving the occasional group recommendation problem. For MoSAN[58], in Section 3.1 of its original paper, we find that it still needs the ad-hoc group for the training. Therefore, it is not a pure, unsupervised group recommendation method. And for the COM [73], it also uses the group assignment information. Besides, for PIT [39], thanks for your suggestion; we missed this paper in the survey process since it seems old. It solves the group recommendation problem via the personal impact topic modeling. And following your suggestion, we will add it to the related work part in the revised paper. For the additional comparison experiments, we think it is hard to produce the results since all these methods are close-resourced. Besides, they are also not new papers and may not achieve very promising performance, especially in the pure, unsupervised learning setting.

## 7.10 Imbalance Problem

Imbalance problem is an essential factor in our scenario. We conduct some case studies in our application. Concretely, we select some hot livestream rooms for the case studies, e.g., the gold-selling livestream room and the grocery-selling livestream room. For the gold-selling livestream room, the imbalance problem exists since most users in this room are interested in the gold and will form one large cluster, but the other users in this room are interested in other items and will form several small clusters. Differently, for the grocery-selling livestream room, the imbalance problem is

not very serious since there are so many small user groups, and the large user group may not easily form. For the downstream task, in our scenario, the imbalance groups may not be a bad phenomenon since we can use this property to find the major user groups in the livestream room and provide recommendations for them.

