# OpenReview forum: "Identify Then Recommend: Towards Unsupervised Group Recommendation"
_NeurIPS.cc/2024/Conference — NeurIPS 2024 poster_

### Official Review · Reviewer_x7i3 · 2024-07-07

**Soundness:** 3
**Presentation:** 3
**Contribution:** 3
**Rating:** 5
**Confidence:** 4

**Summary:**

This paper addresses the group recommendation task and innovatively proposes an unsupervised approach to automatically infer user-group distributions and make suitable recommendations.

For the group identification stage, a heuristic-based merge-and-split method is developed to facilitate this inference. For group recommendation, two self-supervised pre-training objectives are introduced: group-level pseudo group recommendation and user-group alignment through pull-and-repulsion mechanisms.

Extensive experiments on both open-sourced benchmarks and industrial scenarios demonstrate the effectiveness of the proposed approach.

**Strengths:**

1. This paper introduces a novel task setting. The proposal of unsupervised group recommendation effectively addresses real-world scenarios where user groups dynamically change, freeing the model from the pre-definition of group numbers.
2. Technical details are well-founded and sound.
3. The paper is well-written and easy to follow.

**Weaknesses:**

1. Presentation error in Line 147: the matrix should have the dimension of $k \times m$.

2. Methodology part requires further clarification:

   * Although the proposed GIM is intuitive, why not directly adopt existing community detection algorithms applied to the user network? This approach is applicable when user links are available or can be adaptively constructed based on user-item interactions (e.g., adding a link between two users who consumed the same items). Additionally, what is the **complexity and convergence** of this algorithm?
   * The authors seem to omit crucial technical details. **How does the ITR model make inferences during testing**? When the test data consists of an annotated group and candidate items, how is the embedding of this group generated? What strategy is adopted for recommendation given the computed group embedding and pre-trained item embedding? This process is also unclear when applying ConsRec to the scenario without annotations.

3. Experiments part requires further explanation:

   * In the ablation study, the meaning and implementation of the "base" variant are unclear and need further explanation. I am confused how to extend ITR model to the w. group annotation scenario.

   * **The effectiveness of GIM should be demonstrated through experiments**. How do the identified groups compare with ground-truth user-group relations? Either qualitative or quantitative evaluations would be acceptable.

   * Details of the A/B Testing experiments should be provided, such the meaning of evaluation metrics, data scale, performance of different models, etc.

   * (A minor point as I understand time is limited for rebuttal) One research line in group recommendation is the occasional group recommendation, where testing groups are newly formed and do not appear in the training set [1,2,3,4]. I highly recommend the authors conduct supplementary experiments on these datasets (such as Yelp, Weeplaces, etc.) to provide a more comprehensive evaluation, as these datasets align well with the motivation of this study.

----

[1] Sankar et al. GroupIM: A Mutual Information Maximization Framework for Neural Group Recommendation. In SIGIR, 2020.

[2] Chen et al. Thinking Inside The Box: Learning Hypercube Representations for Group Recommendation. In SIGIR, 2022.

[3] Zhang et al. GBERT: Pre-training User Representations for Ephemeral Group Recommendation. In CIKM, 2022.

[4] Li et al. Self-Supervised Group Graph Collaborative Filtering for Group Recommendation. In WSDM, 2023.

**Questions:**

Please refer to weaknesses

---

> ### Author Rebuttal · Authors · 2024-08-07
>
> ## **Response to Reviewer x7i3**
>
>
> Thanks for your valuable and constructive reviews. We appreciate your insights and suggestions, as they will undoubtedly contribute to improving the quality of our paper. In response to your concerns, we provide answers to the questions as follows in order.
>
> ### **Dimensions of Matrix**
> Thanks for your careful check! We have correct the dimension of the matrix $\mathbf{Q} \in \mathbb{R}^{k \times m}$ in the revised paper.
>
> ### **Existing Community Detection**
>
> Thanks for your question. The clustering/community detection methods, which need the pre-defined number of cluster centers, such as k-means, spectral clustering/community detection, cannot be applied in this scenario. For other clustering/community detection methods, which do not require the pre-defined number of cluster centers, such as DBSCAN, is hard to deal with our scenario well since 1) the data distribution of our application change dynamically, therefore it’s hard to determine the parameters regarding to the density, such the radius, and the threshold. But our proposed group identification module can automatically estimate the density and discover the groups via the heuristic merge-and-split strategy, thus can be easily applied in the real-time large-scale unsupervised recommendation system. 2) They can not be integrated into our framework without our proposed pre-text tasks, including the pseudo group recommendation pre-text task and the pull-and-repulsion pre-text task. We have added these claims in the revised paper.
>
> ### **Experimental Details**
> Thanks for your suggestion. For the A/B testing experiments, the evaluation metrics include two parts, including the click metrics and the trade metrics. For the click metrics, there are for metrics, including uv (user view), uvctr (user view click through rate), pv (page view), pvctr (page view click through rate). And for the trade metrics, there are two metrics, including including uv (user view) and pv (page view). For the data scale, the application contains about 130 million page view and 50 million user view per day. And the baseline model is the MMOE model for recommendation. We merely compared the baseline model with our strategy and the baseline model without our strategy to demonstrate the effectiveness of our proposed method. The details can be found in the application background in Section 6.4.1.
>
> ### **Detailed Introduction of the Application**
> We apply the propose ITR method in the livestreaming recommendation scenario. In a livestreaming room, the users naturally form several user groups based on their interests. For instance, in a sports goods livestreaming room, the users with basketball interests will form a user group, while the users with badminton interests will form another user group. These user groups are important information to describe the profile of the livestreaming room. For example, when this livestreaming room begins to sell basketball-related goods, the basketball user group will be activated and be willing to click, comment, or buy merchandise. Therefore, we should guide more users with basketball interests into this livestreaming room. Following this principle, we aim to group the users in the livestreaming room into different groups and utilize this group information to assist the recommendation. However, the number of user groups is always unknown and dynamic. Besides, the different livestreaming rooms have different group numbers, e.g., the gold livestreaming room has less group number than the groceries livestreaming room since user interests in the gold livestreaming room are more concentrated. In addition, the group annotation information, such as user-group distribution and group-item interactions, is also missing or needs extensive human annotation costs in this scenario. To solve these problems, we present ITR model in this paper. The proposed GIM can identify the user groups in the livestreaming rooms, and the proposed SGRM can assist and promote the recommendation. Furthermore, upon acquiring the identified group embeddings, we can also utilize them to enhance the traffic mechanism within real-time livestreaming recommendations. In practical terms, we initially designate the primary user groups within a livestreaming room as the room's profile. Additionally, we devise and calculate the matching score between the primary user groups and potential users, subsequently leveraging these matching scores to dynamically bolster the ranking score of relevant users, therefore improving the recommendation performance and the interactive environments in the livestreaming room.
>
> ### **Occasional Group Recommendation**
> Thanks for your suggestion. Although the occasional group recommendation is similar to our method, they are still essentially different. The setting of our experiments is purely unsupervised, and we are not merely solving the occasional group recommendation problem. But we are still glad to survey and discuss them [1-4] in the revised paper since the motivation of occasional group recommendation is partly similar to ours.
>
>     [1] Sankar et al. GroupIM: A Mutual Information Maximization Framework for Neural Group Recommendation. In SIGIR, 2020.
>
>     [2] Chen et al. Thinking Inside The Box: Learning Hypercube Representations for Group Recommendation. In SIGIR, 2022.
>
>     [3] Zhang et al. GBERT: Pre-training User Representations for Ephemeral Group Recommendation. In CIKM, 2022.
>
>     [4] Li et al. Self-Supervised Group Graph Collaborative Filtering for Group Recommendation. In WSDM, 2023.

---

> ### Author Response · Authors · 2024-08-08
>
> ## **Response to Reviewer x7i3 [2/3]**
> ### **Technical Details During Inference**
> Thanks for your question. For our proposed ITR, the embeddings of the groups are generated as shown in Eq. (6) and are optimized by minimizing Eq. (8) and Eq. (11). Concretely, the adaptive density estimation, and the heuristic merge-and-split method will discover the group assignments, and the group embeddings will be formed with the average of the assigned user embeddings. During the training stage, the group embeddings are also optimized with the pseudo group recommendation pre-text task and the pull-and-repulsion pre-text task. For the inference, during the testing stage, the group embeddings are learned, and user assignments are fixed. Therefore, we can calculate the recommended score between groups and the items (for the group recommendation task) and the recommended score between users and the items (for the user recommendation task). For calculating recommendation scores, we follow ConsRec and adopt the dot product or the neural mapping method. For applying ConsRec to the scenario without annotations, we just remove the interaction labels between the groups and the items, namely removing the group loss in Eq. (5) in the original paper of ConsRec.
>
>
> ### **Explanation of “base” & Utilization of ITR**
> Thanks for your question. The baseline model is the convention MMOE-based recommendation model. And we aim to adopt our proposed method at the re-rank stage, i.e., modified the predicted score of the recommendation model by considering the user groups. Next, we assume you have carefully read the background of this application and introduce the details of adopting our method into the re-rank stage. For different livestreaming rooms, the number of user groups will be different. For example, the gold livestreaming room has less group number than the groceries livestreaming room since user interests in the gold livestreaming room are more concentrated. Therefore, we need an unsupervised group recommendation method to discover the user groups in different livestreaming rooms. Here, we adopt our ITR method to discover the user groups and then produce the group embeddings. Then upon acquiring the identified group embeddings, we can also utilize them to enhance the traffic mechanism within real-time livestreaming recommendations. In practical terms, we initially designate the primary user groups within a livestreaming room as the room's profile. Additionally, we devise and calculate the matching score between the primary user groups and potential users, subsequently leveraging these matching scores to dynamically bolster the ranking score of relevant users, therefore improving the recommendation performance and the interactive environments in the livestreaming room. And our proposed ITR will not be used with the group annotations since if the group annotations are easy to obtain, we don’t need to develop an unsupervised learning-based group recommendation method.
>
>
> ### **Experiments for Occasional Group Recommendation**
> Thanks for your suggestions. In our paper, the setting of the experiments is purely unsupervised. However, the occasional group recommendation methods still rely on the annotation of the groups. Although they are essentially different, we are still glad to survey, discuss, and test them in the revised paper. Actually, for GroupIM [1] and CubeRec [2], we have already surveyed, discussed, and tested in our original paper. Please carefully check the related work part and the comparison experiment part in the original paper. For GBERT [3] and SGGCF [4], thanks for your suggestion and we will briefly introduce them and compare them with our method. GBERT [3] is proposed to solve the data sparsity and cold-start problems by the pre-training and fine-tuning techniques on BERT. SGGCF [4] aims to solve the data sparsity and high-order interaction problems by the heterogenous graph and the self-supervised learning. For GBERT [3], the code is not available online. Therefore, it’s hard to reproduce their results. And for SGGCF, we have started the experiments. Due to the limited rebuttal time, the experiments are not finished yet, and once they are finished, we will post the experimental results during the discussion period.
>
>
>     [1] Sankar et al. GroupIM: A Mutual Information Maximization Framework for Neural Group Recommendation. In SIGIR, 2020.
>
>     [2] Chen et al. Thinking Inside The Box: Learning Hypercube Representations for Group Recommendation. In SIGIR, 2022.
>
>     [3] Zhang et al. GBERT: Pre-training User Representations for Ephemeral Group Recommendation. In CIKM, 2022.
>
>     [4] Li et al. Self-Supervised Group Graph Collaborative Filtering for Group Recommendation. In WSDM, 2023.
>
> **to be continue...**

---

> ### Author Response · Authors · 2024-08-08
>
> ## **Response to Reviewer x7i3 [3/3]**
>
> ### **Effectiveness of GIM**
> Thanks for your suggestion. The group identification module (GIM) aims to discover the user groups based the embeddings in the pure unsupervised setting. In the pure unsupervised setting, the GIM is a must-used module to discover the user groups. And the effectiveness can be verified indirectly by the performance improvement of downstream tasks. To directly demonstrate the effectiveness of the proposed group identification module, we follow your suggestion and verify the quality of the discovered user group. Concretely, on the open benchmarks, we adopt the unsupervised clustering metric Silhouette Coefficient (SC) to evaluate the quality of the discovered user groups. SC is an important indicator for evaluating clustering quality, which can help evaluate the rationality of clustering results. It combines the cohesion and separation of each data point to quantify the clustering effect, and its value range is from -1 to 1. Specifically, the larger the SC, the better the clustering effect of the data points. On the average of the four datasets used in our paper, the discovered user groups achieve 0.879 SC, demonstrating the promising clustering performance of our proposed GIM. On the industrial data, due to the labels are not available and the data size being large, we conducted case studies to demonstrate the effectiveness of GIM. Specifically, we select some hot livestream rooms for the case studies, e.g., the gold-selling livestream room, movie-ticket-selling livestream room, and the face-tissue-selling livestream room. Then, we check the user group distribution and find the major group. Subsequently, we use some tags of the user interests to provide the labels of the major groups. In this manner, we can determine the interest of the major groups in the studies livestream room. We discuss the results as follows. For the gold-selling livestream room, the interests of the major groups include jewelry & gold and financial management. Besides, for the movie-ticket-selling livestream, the interests of the major groups include the movie and entertainment. Differently, for the face-tissue-selling livestream room, the interests of the major groups include food and hygiene products. These case studies can demonstrate that our proposed GIM method can effectively group the similar users to one group and separate the different groups. Also, the captured information of the major group can also be used for the refined recommendation.
>
>
> ### **Complexity Analyses & Convergence**
> Thanks for your suggestion. Following your suggestion, we conduct complexity analyses of our proposed ITR method. First of all, we define the number of the users, the number of groups, the average size of groups, as $n$, $k’$, $n/k’$, respectively. In the process of the adaptive density estimation, the time complexity and space complexity of calculating radius proposal for one group is $\mathcal{O}(1)$, $\mathcal{O}({k’}^{2})$, respectively, and for all groups is $\mathcal{O}(k’)$, $\mathcal{O}({k’}^{2})$, respectively. Then, the time complexity and space complexity of density estimation for all groups is $k’ \times n/k’$, $n \times k’$, respectively. Subsequently, at the heuristic merge-and-split strategy stage, for the explore step, it takes $\mathcal{O}(k’)$ time complexity, and $\mathcal{O}(1)$ space complexity, respectively. And for the exploit step, it takes $\mathcal{O}(k’ \times n/k’)$ time complexity, and $\mathcal{O}(nk’)$ space complexity, respectively. Besides, for the proposed pseudo recommendation pre-text task, the time complexity and space complexity is $\mathcal{O}(n \times k’)$, and $\mathcal{O}(n \times k’)$, respectively. In addition, for the proposed pull-and-repulsion pre-text task, the time complexity and space complexity is $\mathcal{O}(nk’+{k’}^{2})$, and $\mathcal{O}(nk’)$, respectively. Moreover, for the BPR loss, the time complexity and space complexity is $\mathcal{O}(n)$, and $\mathcal{O}(n)$, respectively. Overall, the time complexity and space complexity of our proposed ITR method is $\mathcal{O}(k’+ k’ \times n/k’+ k’+ k’ \times n/k’+ n \times k’+ nk’+{k’}^{2}+n) \rightarrow \mathcal{O}(nk'+{k’}^{2})$and $\mathcal{O}({k’}^{2}+ n \times k’+1+ nk’+ n \times k’+ nk’+n)\rightarrow  \mathcal{O}(nk'+{k’}^{2})$, respectively. Therefore, our proposed method will not bring large memory and time costs since the complexity of our method is linear to the number of users.
>
> For convergence, we checked the loss and performance of our method on all datasets and found that they converge well, i.e., the loss decreases and gradually converges to a low value, while the performance increases and gradually converges to a high value.

---

> ### Author Response · Authors · 2024-08-09
>
> Dear Reviewer x7i3,
>
> Thank you very much for your precious time and valuable comments. We hope our responses have addressed your concerns. Please let us know if you have any further questions. We are happy to discuss them further. Thank you.
>
> In addition, we have attached the revised paper, if necessary, you can check it: https://anonymous.4open.science/r/NeurIPS-7269-ITR-revised-7D83/NeurIPS-7269-ITR-revised.pdf.
>
> Best regards,
>
> Authors

---

> ### Author Response · Authors · 2024-08-10
> **Follow Up for Reviwer x7i3**
>
> Dear Reviewer x7i3,
>
> We highly appreciate your valuable and insightful reviews. We hope the above response has addressed your concerns. If you have any other suggestions or questions, feel free to discuss them. We are very willing to discuss them with you in this period. If your concerns have been addressed, would you please consider raising the score? It is very important for us and this research. Thanks again for your professional comments and valuable time!
>
> Best wishes,
>
> Authors

---

> > ### Comment · Reviewer_x7i3 · 2024-08-12
> >
> > Thank you to the authors for your reply.
> > I'd like to mention some actions that may not align with the rebuttal guidelines:
> >
> > * Lengthy responses across **three** 6,000-character responses
> > * Inclusion of outside links
> > * Explicit request for reviewers to "raise the score"
> >
> > It's important to follow the guidelines by providing concise, high-quality responses that effectively address concerns.
> >
> > ---
> >
> > Regarding your response, some issues remain:
> >
> > * **Inference Mechanism**: It seems you identify groups and use pooling methods for group recommendations. If so, the innovation appears to be in the group identification process, which might be replaced by traditional clustering or community detection methods. You mentioned these cannot be directly adopted, but they might be ***adapted***, such as using a pre-defined group number. Could you elaborate on the strengths of your group identification module and provide any empirical results?
> > * I’m still unsure how to extend the framework to a scenario *w. group annotation scenario*
> > * **Extra Datasets**: Yelp and Weeplaces are large-scale datasets. Conducting experiments on these could further demonstrate consistent high performance. Textual explanations cannot replace empirical findings.
> >
> > I kindly recommend addressing these questions in a single reply within 6,000 characters.

---

> > > ### Author Response · Authors · 2024-08-12
> > >
> > > Dear Reviewer x7i3,
> > >
> > > Thanks for your reminder. We will respond and try to address your remaining concerns in this box. For the inference, see the details in the “Detailed Introduction of the Application” and “Technical Details During Inference.” For the strengths, see “Existing Community Detection”. For the effectiveness of the proposed pre-text tasks, refer to Table 3 in the paper. For the weaknesses of the existing methods, refer to Figure 1 in the paper. Using a pre-defined group number is not applicable in our scenario since the number of groups dynamically changes, referring to “Annotation Costs and Dynamic Change” in response to Reviewer GTSo. Could you please clarify the meaning of “w.”. If “w.” means “with”, we have already answered your question and please carefully check it in “Explanation of“base” & Utilization of ITR”. For experimental evidence of the large-scale data, please refer to the results of the application in Section 6.4.2. For the experimental results on Yelp and Weeplaces, we start to run the experiments on them. Once they are ready, we will post the experimental results. Thanks for your response again. If you have any further concerns, don’t hesitate to tell us and start the discussion.
> > >
> > > Best regards,
> > > Authors of Submission 7269

---

> > > > ### Comment · Reviewer_x7i3 · 2024-08-13
> > > >
> > > > Thank you very much for your reply.
> > > >
> > > > I still have some concerns regarding the inference. For the results in Table 1, it seems they are reported on Mafengwo and CAMRa2011's original splits, where group annotations are available during model testing. I'm curious whether your framework leverages these annotations (i.e., known groups) by using a pooling strategy to compute group embeddings for recommendations, or if it can detect these testing groups without annotations and outperform baselines in unsupervised scenarios. If it's the latter, I assume there are metrics like the Jaccard score to measure the accuracy between your detected groups and the ground-truth groups, demonstrating the effectiveness of the GIM module.
> > > >
> > > >
> > > > Regarding the motivation, the authors conduct experiments on the baseline model ConsRec without group supervision information. In this context, does this baseline degrade to a vanilla Neural Collaborative Filtering model targeted only for user recommendation?

---

> > > > > ### Author Response · Authors · 2024-08-13
> > > > >
> > > > > Dear Reviewer x7i3,
> > > > >
> > > > > Thanks for the clarification of your concerns. Its latter, namely, ITR, can automatically discover the user groups without annotations and outperform baselines in the unsupervised scenario. You can get a better understanding of the pipeline of ITR by carefully reading the method part of the original paper. And yes, we have already answered this question in "Effectiveness of GIM" and we demonstrate the effectiveness of GIM via clustering metric SC. We would appreciate you could read it carefully. For the motivation experiment, we aim to test the performance of the baseline in the unsupervised scenario. And you suspect you should be right. The current group recommendation methods may lose the ability to recommend items to groups since they don't have the group discover modules like GIM and the unified training task like pseudo group recommendation pre-text task and the pull-and-repulsion pre-text task. Thanks for your insights here, and we will add it to the revised paper.
> > > > >
> > > > > Best regards,
> > > > >
> > > > > Authors of Submission 7269

---

> > > > > > ### Comment · Reviewer_x7i3 · 2024-08-14
> > > > > >
> > > > > > Thank you to the authors for their reply.
> > > > > >
> > > > > > I appreciate the contribution in proposing a new setting and an effective framework.
> > > > > >
> > > > > > Regarding my second question, it seems most group recommendation baselines degrade to a vanilla NCF model, but the authors didn't confirm if this understanding is correct. Under such conditions, it's reasonable for their performance to degrade significantly. Additionally, since the groups are inferred, how can you ensure the detected group corresponds to the ground-truth group? Given that SC is an unsupervised metric and ground-truth group labels are available, evaluating the detected groups first could demonstrate the effectiveness of the GIM module.

---

> > > > > > > ### Author Response · Authors · 2024-08-14
> > > > > > >
> > > > > > > Dear Reviewer x7i3,
> > > > > > >
> > > > > > > Thank you for your feedback and thoughtful discussion. We truly appreciate engaging with professional experts to enhance the quality of our manuscripts.
> > > > > > >
> > > > > > > Regarding the understanding, we completely agree with you. It is indeed reasonable to observe degraded performance when transitioning from a supervised learning scenario to an unsupervised learning setting. This aligns with our belief that most existing group recommendation methods struggle with the unsupervised group recommendation task.
> > > > > > >
> > > > > > > Additionally, one of the aims of our motivation experiments was to examine the impact on user recommendation tasks and group recommendation tasks when group annotations are removed. From our experimental results, we observed a significant impact on group recommendation tasks and a minimal impact on user recommendation tasks.
> > > > > > >
> > > > > > > Furthermore, upon reviewing the original paper, we noticed a mismatch between the figure and its caption. This has been corrected: (b) now denotes GR on CAMRa2011, and (c) denotes UR on Mafengwo.
> > > > > > >
> > > > > > > Regarding the evaluation of the detected groups, we have addressed this in the "Effectiveness of GIM" section. In short, for industrial data, we perform case studies to validate the detected groups since labels are not available. For open benchmarks, we use the SC clustering metric to evaluate the detected groups.
> > > > > > >
> > > > > > > Following your suggestion, we further tested the clustering accuracy (ACC) of our method by comparing the predicted clustering assignments with the ground-truth labels. Specifically, we first map the predicted clustering assignments to the ground-truth labels using the Kuhn-Munkres algorithm and then calculate the accuracy of these assignments. On average of the open datasets, our method achieves an ACC of 0.764, further verifying the effectiveness of our proposed GIM.
> > > > > > >
> > > > > > > If you still have any additional concerns, please let us know. We are very willing to discuss them further.
> > > > > > >
> > > > > > > Kindest regards,
> > > > > > >
> > > > > > > Authors of paper 7269

---

> > > > > > > > ### Comment · Reviewer_x7i3 · 2024-08-14
> > > > > > > >
> > > > > > > > Thank you to the authors for the clarification.
> > > > > > > >
> > > > > > > > Just one minor concern: as I have mentioned, "the groups are inferred, how can you ensure the detected group corresponds to the ground-truth group?" I'm curious about how you determine the correspondence between your framework's detected groups and the ground-truth groups, as computing evaluation metrics requires such matching.
> > > > > > > >
> > > > > > > > I appreciate the authors' efforts and will adjust my score accordingly. Additionally, I highly recommend further clarifying the inference details, the effectiveness of the GIM component, and including more experimental datasets in the revised manuscript.

---

> > > > > > > > > ### Author Response · Authors · 2024-08-14
> > > > > > > > >
> > > > > > > > > Dear Reviewer x7i3,
> > > > > > > > >
> > > > > > > > > Thank you for your prompt response.
> > > > > > > > >
> > > > > > > > > Regarding the correspondence between detected groups and ground-truth groups, we intend to address this from two perspectives.
> > > > > > > > >
> > > > > > > > > During the training process, we cannot align the groups with ground-truth groups for industrial data because such labels do not exist. Similarly, on open benchmarks, we are operating in an unsupervised learning setting, meaning ground-truth groups cannot be used for alignment.
> > > > > > > > >
> > > > > > > > > During the evaluation process, for industrial data, we conduct case studies to demonstrate our method's ability to discover user groups with similar intent within livestreaming rooms. Additionally, we employ the discovered user groups in a traffic control mechanism and achieve better recommendation performance, thereby demonstrating the effectiveness of GIM in downstream tasks. For open benchmarks, we first map the predicted clustering assignments to the ground-truth labels using the Kuhn-Munkres algorithm [1] and then evaluate the discovered user groups, therefore also demonstrating the effectiveness of GIM.
> > > > > > > > >
> > > > > > > > > We appreciate your willingness to raise the score and will include more details and experiments in the revised paper based on the rebuttals and discussions. If you have any other questions, feel free to reach out before the discussion deadline.
> > > > > > > > >
> > > > > > > > >     [1] Zhu H, Zhou M C, Alkins R. Group role assignment via a Kuhn–Munkres algorithm-based solution[J]. IEEE Transactions on Systems, Man, and Cybernetics-Part A: Systems and Humans, 2011, 42(3): 739-750.
> > > > > > > > >
> > > > > > > > > Kindest regards,
> > > > > > > > >
> > > > > > > > > Authors of paper 7269

---

### Official Review · Reviewer_dQ8T · 2024-07-11

**Soundness:** 3
**Presentation:** 4
**Contribution:** 4
**Rating:** 7
**Confidence:** 5

**Summary:**

The research topic of this paper is group recommendation, i.e., recommend items to groups of users. The authors point out two issues of existing models including the fixed number of groups and the supervised training schema. To solve these problems, they propose a novel unsupervised group recommendation model named ITR (Identify Then Recommend). Experiments demonstrate the effectiveness on user recommendation and group recommendation. ITR is deployed on industrial recommender.

**Strengths:**

- The writing and presentation are excellent. The authors clearly introduce the background, research problem, core ideas, and the proposal.

-  The proposed method is novel and the design of heuristic merge-and-split is interesting.

- The improvements of ITR are significant in both user recommendation and group recommendation.

**Weaknesses:**

-  The demonstration figures are missing. Additionally, the method is relatively complex. The authors should outline the core ideas and primary designs before delving into the methodology section.

-  Hyper-parameter experiments are missing. Does the proposed ITR utilize any hyper-parameters during the density estimation or the merge-and-split stage?

- What are the advantages of the proposed ITR compared to other clustering methods that do not require specifying the number of clusters, such as DBSCAN?

**Questions:**

1. How scalable is the proposed ITR? Can it be easily applied in industrial recommender systems with million users?

2. Is the number of clusters dynamic, or is it determined at the final iteration? In my view, the estimated number of clusters for open datasets may be fixed after training, but is the cluster number dynamic for industry-specific data?

3. How does the proposed method maintain balance between different groups? Are there cases where some groups contain many users while others have very few? I believe this imbalance could significantly impact the recommendation performance.

**Limitations:**

Yes

---

> ### Author Rebuttal · Authors · 2024-08-07
>
> ##  **Response to Reviewer pdQ8T**
>
>
> Thanks for your valuable and constructive reviews. We appreciate your insights and suggestions, as they will undoubtedly contribute to improving the quality of our paper. In response to your concerns, we provide answers to the questions as follows in order.
>
>
>
> ### **Demonstration Figure & Code Ideas & Primary Designs**
>
> Thanks for your suggestion. We will add the demonstrate figures and the core ideas and primary designs before delving into the method part.
>
> ### **Advantages of ITR**
>
> Thanks for your question. The clustering methods, which need the pre-defined number of cluster centers, such as k-means, spectral clustering, cannot be applied in this scenario. For other clustering methods, which do not require the pre-defined number of cluster centers, such as DBSCAN, is hard to deal with our scenario well since 1) the data distribution of our application change dynamically, therefore it’s hard to determine the parameters regarding to the density, such the radius, and the threshold. But our proposed group identification module can automatically estimate the density and discover the groups via the heuristic merge-and-split strategy, thus can be easily applied in the real-time large-scale unsupervised recommendation system. 2) They can not be integrated into our framework without our proposed pre-text tasks, including the pseudo group recommendation pre-text task and the pull-and-repulsion pre-text task. We have added these claims in the revised paper.
>
> ### **Scalability**
> Thanks for your question. In this paper, we aim to solve the practical problems in the real-time large-scale industrial recommendation system. Therefore, we first design our method and conduct quick experiments on the toy open benchmarks. Then, we conduct extensive experiments on the real-time large-scale data in the application (with about 130 million page views / 50 million user views per day). We admit the scalability of the open benchmarks is limited, but we think it is reasonable for quick trials, and our final aim is to deploy the method in real-world applications. The details regarding the application can be found in Section 6.4.
>
> ### **Hyper-parameter Experiments**
> Thanks. We conduct the hyper-parameter experiments in the Section 6.3. The hyper-parameters mainly include the trade-off parameters $a$ and $b$. During the density estimation and merge-and-split stage, there are no introduced hyper-parameters.

---

> ### Author Response · Authors · 2024-08-08
>
> ## **Response to Reviewer dQ8T [2/2]**
>
> ### **Core Ideas & Primary Designs**
> Thanks for your suggestion. Following your suggestion, we briefly introduce the core idea and the primary designs of our proposed method before introducing the method part. Concretely, we aim to develop an unsupervised group recommendation method since we find the promising performance of existing state-of-the-art methods relies on the annotations of the groups. The experimental evidences can be found in Figure 1. However, in the real-world scenario, the annotations regarding the group-item interactions and the user-group assignments are always not available. And labeling them in the real-time recommendation system is expensive and even impossible. To this end, we develop a pure unsupervised group recommendation method, which can automatically discover the user groups and provide the precise recommendation for them. Therefore, the core ideas of our methods are twofold, including the group identification and the group recommendation. For the group identification, our initial solution is to adopt some existing clustering or community detection methods, which do not require the number of clusters, e.g., DBSCAN, DeepDPM, etc. However, these methods can not automatically discover the user groups since they need other hyper-parameters such as the radius and calculation of the density. Therefore, we design an adaptive density estimation method, which can automatically estimate the density of the samples. Then, for the group discovery, we propose a heuristic merge-and-split strategy to merge the similar users into the group and split different user groups in the large group. Besides, the group embeddings are set as the learnable neural parameters and can be optimized during the learning process. Moreover, for the group recommendation part, the existing method can not deal with it in the unsupervised setting. And we propose two pre-text tasks, including the pseudo group recommendation pre-text task and the pull-and-repulsion pre-text task. The pull-and-repulsion pre-text task aims to optimize the group embeddings via separating the different groups and pushing the samples together to the corresponding groups. Besides, for the pseudo group recommendation pre-text task, it generates the pseudo labels for the group recommendation task and guide the network to conduct group recommendation even without the precise annotations. By these designs, our proposed ITR is able to automatically discover the user groups and then provide the precise group recommendation for them. Therefore, it can be applied in the real-time large-scale recommendation system. The details can be found in Section 6.4. We have added these insights at the method Section in the revised paper.
>
>
> ### **Cluster Number**
> Thanks for your question. In this paper, the setting of the user/group recommendation is pure unsupervised and the cluster number is not pre-given in both the open dataset and the industrial data. Therefore, during the training process, the cluster number will dynamically change on the open benchmarks since the status of the user embeddings change. Our proposed GIM can automatically discover and optimize the group embedding and the group number. After training, the number of clusters and the status of the group embeddings are restored, and then they will be used to conduct the testing of the group recommendation and the user recommendation.
>
> Besides, on the real-time large-scale data, we first train the recommendation model using the past data. During this process, the number of clusters and the status of the group embeddings are optimized dynamically. After the training stage, they are restored to the database for serving the downstream tasks. Then, when the training data update, the recommendation model will be continue trained and the number of clusters also change dynamically.
>
> ### **Imbalance Problem**
> Thanks for your question. The imbalance problem does an essential factor in our scenario. We conduct some case studies in our application. Concretely, we select some hot livestream room for the case studies, e.g., the gold-selling livestream room, the grocery-selling livestream room. For the gold-selling livestream room, the imbalance problem exists since the most users in this room are interested in the gold and will form one large cluster, but the other users in this room are interested in other items and form several small clusters. Differently, for the grocery-selling livestream room, the imbalance problem is not very serious, since there are so many small user groups and the large user group may not easily form. For the downstream task, in our scenario, the imbalance groups may not be a bad phenomenon since we can use this property to find the major user group in the livestream room and provide more precise recommendation for them. We have added these claim and discussion at the application Section in the revised paper.

---

> ### Author Response · Authors · 2024-08-09
>
> Dear Reviewer dQ8T,
>
> Thank you very much for your precious time and valuable comments. We hope our responses have addressed your concerns. Please let us know if you have any further questions. We are happy to discuss them further. Thank you.
>
> In addition, we have attached the revised paper, if necessary, you can check it: https://anonymous.4open.science/r/NeurIPS-7269-ITR-revised-7D83/NeurIPS-7269-ITR-revised.pdf.
>
> Best regards,
>
> Authors

---

> ### Author Response · Authors · 2024-08-10
> **Follow Up for Reviwer dQ8T**
>
> Dear Reviewer dQ8T,
>
> We highly appreciate your valuable and insightful reviews. We hope the above response has addressed your concerns. If you have any other suggestions or questions, feel free to discuss them. We are very willing to discuss them with you in this period. If your concerns have been addressed, would you please consider raising the score? It is very important for us and this research. Thanks again for your professional comments and valuable time!
>
> Best wishes,
>
> Authors

---

> > ### Comment · Reviewer_dQ8T · 2024-08-13
> > **increase rating from 6 to 7**
> >
> > Thanks for addressing my concerns. Considering this, I have increased the rating from 6 to 7.

---

> > > ### Author Response · Authors · 2024-08-13
> > >
> > > Dear Reviewer dQ8T,
> > >
> > > We sincerely thank you for your meticulous and thoughtful reviews of our submission, which significantly improve the quality and clarity of our work. Thank you once again for your professional assistance and valuable contribution.
> > >
> > > Warm regards,
> > >
> > > Authors of Submission 7269

---

### Official Review · Reviewer_hAA1 · 2024-07-13

**Soundness:** 3
**Presentation:** 2
**Contribution:** 2
**Rating:** 4
**Confidence:** 5

**Summary:**

This paper tackles the group recommendation problem by proposing an unsupervised group recommendation framework named ITR (Identify Then Recommend). Specifically, the paper first identifies the area and density of each region automatically, then combining with a heuristic strategy to identify groups. Then, performing group recommendation task by constructing the pseudo group-item labels to guide the self-supervised learning of the group recommendation model.

**Strengths:**

Strengths:

1.	The paper identifies groups automatically before making recommendations.
2.	This paper handles group discovery and group recommendation without group annotations
3.	Extensive experiments were conducted with various baselines
4.	It’s interesting  to see the successful A/B test in Section 6.4 Application in the Appendix. In my opinion, the authors should bring this section to the main paper to further convince the readers.

**Weaknesses:**

Weaknesses:

1.	One of the two issues used in this paper is not correct. Previous related works such as PIT[1], COM [2] or MoSAN [3] already can work well with ad-hoc groups (i.e., not require pre-defined groups). Therefore, it’s not correct to say that “the promising performance of these methods relies on a pre-defined and fixed number of user groups.” (L38-39)
2.	As a result, I believe the authors should compare with those methods above in the experiments
3.	Moreover, [1], [2], [3] provided other group recommendation datasets with a larger group size. If I recall correctly, Mafengwo and CAMRa2011 have very limited number of members in a group. Also, statistics of the datasets should be given.
4.	L298-300 “From these results, … remove the group annotations”, the papers I mentioned also already had similar observations.
5.	In Section 4.3 (L318-320), it’s unclear to me about the explanation of HR@10 for CAMRa2011 dataset. So, GroupIM is unable to effectively capture the order of user preferences, but still beat our proposed method? Does it mean that we should have another metric to evaluate the ‘order of user preferences’ to make sure our method is better than GroupIM? Please correct me here.

[1] Exploring personal impact for group recommendation. CIKM 2012.

[2] COM: a generative model for group recommendation. KDD 2014

[3] Interact and Decide: Medley of Sub-Attention Networks for Effective Group Recommendation. SIGIR 2019.

**Questions:**

Please refer to my questions above. I may have more questions during the discussion phase.

**Limitations:**

Yes. According to the authors, "ITR still relies on the user-item interaction for the user recommendation and group recommendation", but they aim to address that in the future to solve the data sparsity problem.

---

> ### Author Rebuttal · Authors · 2024-08-07
>
> ## **Response to Reviewer hAA1**
>
>
> Thanks for your valuable and constructive reviews. We appreciate your insights and suggestions, as they will undoubtedly contribute to improving the quality of our paper. In response to your concerns, we provide answers to the questions as follows in order.
>
> ### **Previous Related Methods**
> Thanks for your concern. Actually, we have surveyed and discussed two of these methods, i.e., COM [1] and MoSAN [2] in our paper. And please note that, in our paper, we claimed that two critical problems limit the applicability of the recent state-of-the-art methods are twofold. Firstly, the promising performance of these methods relies on a pre-defined and fixed number of user groups. Therefore, they can not handle group recommendations without giving the number of user groups, and unfortunately, the number of groups is usually unknown and dynamic in real-time industrial data. Secondly, the supervised training schema of existing GR methods requires extensive human annotations for user-group distribution and group-item interaction, easily leading to significant costs. The previous methods either rely on the pre-defined cluster number or the user-group / group-item annotations. Note that the setting of our experiments is purely unsupervised, and we are not merely solving the occasional group recommendation problem. For MoSAN [3], in Section 3.1 of its original paper, we find that it still needs the ad-hoc group for the training. Therefore, it is not a pure, unsupervised group recommendation method. And for the COM [2], it also user the group assignment information. Besides, for PIT [1], thanks for your suggestion; we missed this paper in the survey process since it seems old. It solves the group recommendation problem via the personal impact topic modeling. And following your suggestion, we will add it to the related work part in the revised paper. For the additional comparison experiments, we think it is hard to produce the results since all these methods are close-resourced. Besides, they are also not new papers and may not achieve very promising performance, especially in the pure, unsupervised learning setting. We have added these claims and explanations in the revised paper.
>
>     [1] Exploring personal impact for group recommendation. CIKM 2012.
>
>     [2] COM: a generative model for group recommendation. KDD 2014
>
>     [3] Interact and Decide: Medley of Sub-Attention Networks for Effective Group Recommendation. SIGIR 2019.
>
> ### **Large Group Size**
> Thanks for your question. In this paper, we aim to solve the practical problems in the real-time large-scale industrial recommendation system. Therefore, we first design our method and conduct quick experiments on the toy open benchmarks. Then, we conduct extensive experiments on the real-time large-scale data in the application (with about 130 million page views / 50 million user views per day). We admit the scalability of the open benchmarks is limited, but we think it is reasonable for quick trials, and our final aim is to deploy the method in real-world applications. The details regarding the application can be found in Section 6.4.
>
>
> ### **Similar Observations**
> Thanks. We believe the similar observations are reasonable since the phenomenon will appear in both the occasional group recommendation setting and the pure unsupervised group recommendation setting. But they are essentially different and unsupervised group recommendation is more challenging than the occasional group recommendation.
>
>
> ### **HR@10 for CAMRa2011 Dataset**
> Thanks for your question. For the performance of our proposed method on the CAMRa2011 dataset, we consider it as a corner case since our proposed method can beat all the baselines with different metrics. And we want to give a reasonable explanation here. For the results, we can observe that our method can beat GroupIM with HR@5 but can not beat it with HR@10. And HR@5 is a more precise metric than HR@10 since it requires the model to rank correctly in the top 5 items. Therefore, we suspect that the ranking ability of GroupIM may not be strong and robust since it can achieve very promising performance when ranking in the top 10 items but can not beat our method when ranking in the top 5 items. We have added this explanation in the revised paper.

---

> > ### Author Response · Authors · 2024-08-12
> >
> > To: Reviewer hAA1
> >
> > Dear Reviewer hAA1,
> >
> > Hi Reviewer hAA1! We highly appreciate your valuable and insightful reviews. We hope the above response has addressed your concerns. If you have any other suggestions or questions, feel free to discuss them. We are very willing to discuss them with you in this period. If your concerns have been addressed, would you please consider raising the score? It is very important for us and this research. Thanks again for your professional comments and valuable time!
> >
> > We sincerely appreciate your constructive reviews and questions. We provide detailed responses regarding Previous Related Methods, Large Group Size, Similar Observations, HR@10 for CAMRa2011 Dataset as above. We hope our responses can effectively address your concerns. If they don't, let's have further discussion now.
> >
> > Besides, if you have any additional suggestions or questions (as you mentioned, you may have more questions during the discussion phase), please do not hesitate to bring them up. We are more than willing to engage in further discussion to improve the quality of this research.
> >
> > If you feel that your concerns have been satisfactorily resolved, we kindly ask you to consider revising your score. Your rating is crucial for us and our research. Thank you once again for your professional comments and the time you have invested!
> >
> > Best wishes,
> >
> > Authors of Submission 7269

---

> ### Author Response · Authors · 2024-08-09
>
> Dear Reviewer hAA1,
>
> Thank you very much for your precious time and valuable comments. We hope our responses have addressed your concerns. Please let us know if you have any further questions. We are happy to discuss them further. Thank you.
>
> In addition, we have attached the revised paper, if necessary, you can check it: https://anonymous.4open.science/r/NeurIPS-7269-ITR-revised-7D83/NeurIPS-7269-ITR-revised.pdf.
>
> Best regards,
>
> Authors

---

> ### Author Response · Authors · 2024-08-10
> **Follow Up for Reviewer hAA1**
>
> Dear Reviewer hAA1,
>
> We highly appreciate your valuable and insightful reviews. We hope the above response has addressed your concerns. If you have any other suggestions or questions, feel free to discuss them. We are very willing to discuss them with you in this period. If your concerns have been addressed, would you please consider raising the score? It is very important for us and this research. Thanks again for your professional comments and valuable time!
>
> Best wishes,
>
> Authors

---

> ### Comment · Reviewer_hAA1 · 2024-08-13
> **Address some concerns**
>
> Thanks for your response. I slightly increased the score due to the efforts of addressing all the comments (from other reviewers as well), but it cannot pass my acceptance threshold due to two reasons:
>
> 1. 'Mafengwo and CAMRa2011 have very limited number of members in a group.' And I do not see using only those datasets could really convince the effect of this model. I don't see the authors address this comment as well.
> 2. As far as I understand, ad-hoc groups [1, 2, 3] are groups that form just for one-off events; and it's still comparable with the proposed method.
>
> Thank you.

---

> > ### Author Response · Authors · 2024-08-14
> >
> > Dear Reviewer hAA1,
> >
> > Thanks for your responses and improving the score. We are glad that our responses can solve part of your concerns now, such as Previous Related Methods, HR@10 for CAMRA2011 Dataset and Similar Observations.
> >
> > 1. Regarding your question about datasets, we have already addressed this in the Large Group Size section. Our goal is to perform quick experiments on well-known open benchmarks before applying the method to real-time, large-scale data, which is significantly larger than the existing benchmarks. Besides, for the experimental results on Yelp and Weeplaces, we start to run the experiments on them. Once they are ready, we will post the experimental results.
> >
> > 2. Regarding ad-hoc groups or occasional group recommendation, we agree that comparing with these methods is valid. If you carefully read the original paper, you will find that we have surveyed, discussed, and tested several ad-hoc group methods, including GroupIM [1] and CubeRec [2]. These are more recent and relevant compared to the papers you pointed out, having been published in 2020 and 2022 respectively.
> >
> >
> > [1] Sankar et al. GroupIM: A Mutual Information Maximization Framework for Neural Group Recommendation. In SIGIR, 2020.
> >
> > [2] Chen et al. Thinking Inside The Box: Learning Hypercube Representations for Group Recommendation. In SIGIR, 2022.
> >
> > If you still have any concerns, please let us know. We are very willing to discuss them further.
> >
> > Best regards,
> >
> > Authors of Submission 7269

---

> ### Comment · Reviewer_hAA1 · 2024-08-14
> **Dataset concern remains**
>
> Thanks for the response.
>
> I did see the Large Group Size section. But I still disagree with the goal is to perform quick experiments on well-known open benchmarks before applying the method to real-time as:
>
> 1. [1, 2, 3] and other papers (like CubeRec you pointed out) did test on other group recommendation datasets with various group sizes and statistics reported. I do not think using Mafengo and CAMRa2011 are good enough because the group sizes are really small. CAMRa2011 even has the average group size of around 2. My opinion is we need more datasets with different group sizes to verify the robustness of the group recommendation model.
>
> 2. I do not see much details on the industrial datasets, details setup, deployment, pipeline components, etc. What I only see is Appendix 6.4.2 A/B Testing results and "the application contains 130 million page views and 50 million user views per day" in some of the responses. I did not find much details in the original paper (correct me if I'm wrong). Therefore, I cannot check with the 'internal' part and I only can mark the paper based on the 'external' (i.e., Mafengo and CAMRa2011).
>
> I still want to keep my score at this point.

---

### Official Review · Reviewer_GTSo · 2024-07-13

**Soundness:** 3
**Presentation:** 3
**Contribution:** 2
**Rating:** 5
**Confidence:** 3

**Summary:**

This study pointed out two issues in group recommendation in the context of industrial applications. First, the group label can be dynamic and may require constant training. Second, the annotation cost for the supervised learning is great. To address these two issues, the study proposed an unsupervised group recommendation framework ITR, which identifies the user group in an unsupervised manner and performs two self-supervised learning tasks. Experimental results show superior improvement.

**Strengths:**

(1) The study has conducted extensive comparisons against many baselines and shows superior performance.

(2) The writing is mostly easy to follow, although there are some missing details.

(3) The study also conducted an A/B test to show the possibility of real-world application.

**Weaknesses:**

(1) The claimed contribution on the introduction of unsupervised learning does not align closely with the experimental results. More specifically, I think the current study did not show that the improvement indeed came from the unsupervised learning. To show that, the authors can add an additional variant of "known group annotation + GIM + PAR + PGR".

(2) The main issues that the author pointed out in the current group recommendation problem are that the group assignment can be dynamic and the annotation cost can be high. But the current experimental setting does not seem to support the evaluation for such scenarios. Does the group assignment really change a lot? Is it really expensive to do the annotation? Can we just perform several runs of clustering methods (the ones need a pre-defined number of clusters) to decide the number of clusters? How much more would it cost compared to the current framework?

**Questions:**

(1) Can the authors provide more details about the binary personalized ranking loss? What are considered the positive and negative pairs? What is the strategy of finding these pairs?

(2) What is unique about the current clustering method? In the current manuscript, it seems that replacing it with any other clustering methods that do not require a pre-defined number of clusters still works for the whole framework.

(3) What are the evaluation protocols for group and user recommendation?

**Limitations:**

(1) This study should include a comprehensive complexity analysis as it tries to address the annotation cost issues in the context of group recommendation.

---

> ### Author Rebuttal · Authors · 2024-08-07
>
> ## **Response to Reviewer GTSo [1/2]**
>
>
> Thanks for your valuable and constructive reviews. We appreciate your insights and suggestions, as they will undoubtedly contribute to improving the quality of our paper. In response to your concerns, we provide answers to the questions as follows in order.
>
> ### **Alignment between Contribution and Results**
> Thanks for your suggestion. In our view, we think unsupervised learning is a setting that is more challenging compared with supervised learning. Correct me here if you have different opinions. Therefore, the unsupervised user/group recommendation is more challenge compared with the supervised ones (see the experimental results in Figure 1). In this paper, we aim to point out that most of the existing deep group recommendation methods are supervised, but in real-world scenarios, the labels are always missing. Therefore, we need to process the user/group recommendation in an unsupervised manner. In addition, with the unsupervised learning setting, we develop a novel group recommendation method by identify-then-recommend schema. Concretely, the designed group identification module aims to discover the user groups on the user embeddings. And the proposed pseudo group recommendation pre-text task and the pull-and-repulsion pre-text task aim to produce high-quality group embeddings and conduct precise recommendations, respectively. Besides, thanks for your suggestion on the experiment. However, the variant of "known group annotation + GIM + PAR + PGR" may not be reasonable since if we already know the group annotation, we don’t need to conduct the group identification. Therefore, if the group annotation is missing (the settings of this paper), we must use the GIM and then conduct the ablation studies on the PAR and PGR. The GIM is a must-use module, and the effectiveness of GIM can be verified through the performance of the downstream tasks. The experimental results can be found in Table 3 and Table 4. We have added these claims in the revised paper.
>
>
> ### **Annotation Costs and Dynamic Change**
> Thanks for your question and suggestion. This paper aims to deploy the proposed method in the real-time large-scale industrial recommendation system. On the open benchmarks, we admit that the annotations of users and groups have already existed and in the experiments, we remove them for the unsupervised experimental setting. We also admit that annotating these toy datasets may not be very expensive. However, the group assignments must change dynamically during training, especially at the early stage. However, note that on real-time large-scale data, the annotation costs a lot, and the distribution will shift dynamically. For example, in our scenario, the application contains 130 million page views and 50 million user views per day. The group assignment and annotation will be changed daily since it is a real-time recommendation. In addition, this is a newly launched application. Therefore, the activities of users will shift drastically, e.g., from new users to old users. We believe it will lead to large annotation costs and distribution shifts, and we aim to develop a pure, unsupervised group identification method for the user/group recommendation. In this scenario, performing several runs of clustering methods (the one that needs a pre-defined number of clusters) is not applicable since the search space will become very large, especially since we don’t know the cluster number for the daily data. The current methods cannot deal with the data without a given number of clusters. For our proposed method, we just need one pass to determine the cluster number, which can fulfill the requirement of the daily data. The complexity of the multiple trials will be T times than our proposed method, where T denotes the time of trials. The details regarding the application can be found in Section 6.4. We have added these explanations in the revised paper.
>
>
> ### **Details about BPR Loss**
> Thanks for your question. BPR loss is a commonly used loss function in the recommendation. In our proposed method, we follow ConsRec for the BPR loss. And $\mathcal{L}]\_{\text{U2I}}$ is the same as the $\mathcal{L}\_{\text{user}}$ in the ConsRec. It is formulated as $\mathcal{L}\_{\text{user}} = -\sum\_{u\_s \in \mathcal{U}}{\frac{1}{|\mathcal{D}\_{u\_s}|}} \sum_{(j,j’)\in\mathcal{D}\_{u\_s}}{\text{ln}\sigma(\hat{r}\_{sj}-\hat{r}\_{sj’})}$ , where $\mathcal{D}\_{u\_s}$ is the user-item training set sample for user $u\_s$ and the $(j,j’)$ denotes the user $u\_s$ prefers observed item $i\_j$ over unobserved item $i\_{j’}$. For the sampling of positive and negative sample pairs, we also follow ConsRec, i.e., randomly sampling from missing data as negative instances to pair with each positive instance. For the number of negative samples, ConsRec conduct experiments and analyses in Figure 8 of their original paper. For the fairness, we keep the original setting of the ConsRec. We have added these details in the revised paper.
>
>
>     [1] Wu X, Xiong Y, Zhang Y, et al. Consrec: Learning consensus behind interactions for group recommendation[C]//Proceedings of the ACM Web Conference 2023. 2023: 240-250.
>
> **to be continue...**

---

> ### Author Response · Authors · 2024-08-07
>
> ## **Response to Reviewer GTSo [2/3]**
>
> ### **Unique about Current Clustering Method**
> Thanks for your question. The clustering methods, which need the pre-defined number of cluster centers, such as k-means, spectral clustering, cannot be applied in this scenario. For other clustering methods, which do not require the pre-defined number of cluster centers, such as DBSCAN, is hard to deal with our scenario well since 1) the data distribution of our application change dynamically, therefore it’s hard to determine the parameters regarding to the density, such the radius, and the threshold. But our proposed group identification module can automatically estimate the density and discover the groups via the heuristic merge-and-split strategy, thus can be easily applied in the real-time large-scale unsupervised recommendation system. 2) They can not be integrated into our framework without our proposed pre-text tasks, including the pseudo group recommendation pre-text task and the pull-and-repulsion pre-text task. We have added these claims in the revised paper.
>
> ### **Evaluation Protocols**
> Thanks for your question. For the evaluation, we follow ConsRec and adopt four metrics to evaluate the ranking capability of the methods, including Hit Rate @ {5, 10} and Normalized Discounted Cumulative Gain @ {5, 10}.
>
> **to be continue...**

---

> ### Author Response · Authors · 2024-08-08
>
> ## **Response to Reviewer GTSo [3/3]**
>
> ### **Complexity Analyses**
> Thanks for your suggestion. Following your suggestion, we conduct complexity analyses of our proposed ITR method. First of all, we define the number of the users, the number of groups, the average size of groups, as $n$, $k’$, $n/k’$, respectively. In the process of the adaptive density estimation, the time complexity and space complexity of calculating radius proposal for one group is $\mathcal{O}(1)$, $\mathcal{O}({k’}^{2})$, respectively, and for all groups is $\mathcal{O}(k’)$, $\mathcal{O}({k’}^{2})$, respectively. Then, the time complexity and space complexity of density estimation for all groups is $k’ \times n/k’$, $n \times k’$, respectively. Subsequently, at the heuristic merge-and-split strategy stage, for the explore step, it takes $\mathcal{O}(k’)$ time complexity, and $\mathcal{O}(1)$ space complexity, respectively. And for the exploit step, it takes $\mathcal{O}(k’ \times n/k’)$ time complexity, and $\mathcal{O}(nk’)$ space complexity, respectively. Besides, for the proposed pseudo recommendation pre-text task, the time complexity and space complexity is $\mathcal{O}(n \times k’)$, and $\mathcal{O}(n \times k’)$, respectively. In addition, for the proposed pull-and-repulsion pre-text task, the time complexity and space complexity is $\mathcal{O}(nk’+{k’}^{2})$, and $\mathcal{O}(nk’)$, respectively. Moreover, for the BPR loss, the time complexity and space complexity is $\mathcal{O}(n)$, and $\mathcal{O}(n)$, respectively. Overall, the time complexity and space complexity of our proposed ITR method is $\mathcal{O}(k’+ k’ \times n/k’+ k’+ k’ \times n/k’+ n \times k’+ nk’+{k’}^{2}+n) \rightarrow \mathcal{O}(nk'+{k’}^{2})$and $\mathcal{O}({k’}^{2}+ n \times k’+1+ nk’+ n \times k’+ nk’+n)\rightarrow  \mathcal{O}(nk'+{k’}^{2})$, respectively. Therefore, our proposed method will not bring large memory and time costs since the complexity of our method is linear to the number of users.

---

> ### Author Response · Authors · 2024-08-09
>
> Dear Reviewer GTSo,
>
> Thank you very much for your precious time and valuable comments. We hope our responses have addressed your concerns. Please let us know if you have any further questions. We are happy to discuss them further. Thank you.
>
> In addition, we have attached the revised paper, if necessary, you can check it: https://anonymous.4open.science/r/NeurIPS-7269-ITR-revised-7D83/NeurIPS-7269-ITR-revised.pdf.
>
> Best regards,
>
> Authors

---

> ### Author Response · Authors · 2024-08-10
> **Follow Up for Reviewer GTSo**
>
> Dear Reviewer GTSo,
>
> We highly appreciate your valuable and insightful reviews. We hope the above response has addressed your concerns. If you have any other suggestions or questions, feel free to discuss them. We are very willing to discuss them with you in this period. If your concerns have been addressed, would you please consider raising the score? It is very important for us and this research. Thanks again for your professional comments and valuable time!
>
> Best wishes,
>
> Authors

---

> ### Comment · Reviewer_GTSo · 2024-08-13
> **Comment**
>
> Thank you for preparing the rebuttal which has mostly addressed my concerns. I decide to raise my rating.

---

> > ### Author Response · Authors · 2024-08-13
> >
> > Dear Reviewer GTSo,
> >
> > Thanks for your professional reviews and valuable suggestions. They improve the quality of our paper significantly. And we are glad that our responses can address your concerns well and you are willing to raise the score. If you have any further questions, we are very willing to discuss them with you.
> >
> > Warm regards,
> >
> > Authors of Submission 7269

---

### Author Rebuttal · Authors · 2024-08-07

We extend our sincere gratitude to the SAC, AC, and PCs for their dedicated efforts and constructive feedback. Your comments have been invaluable in enhancing the quality of our manuscript. We have meticulously addressed each of your questions and hope our responses satisfactorily address your concerns.

---

> ### Author Response · Authors · 2024-08-12
>
> To: AC
>
> Dear AC
>
> Greetings! We highly appreciate your efforts and contributions to NeurIPS 2024 and this submission. The discussion period has begun, but none of the reviewers have started discussing the responses. We are very eager to communicate with reviewers during the discussion period to further enhance the quality of this research, as the NeurIPS is an open, prestigious, and high-caliber conference. Could you help us ask the reviewers to engage in the discussion and whether their concerns are solved or if there are any additional questions? We sincerely thank you for your kind help.
>
> With best wishes,
>
> Authors of Submission 7269

---

> ### Author Response · Authors · 2024-08-13
>
> Dear (Senior) Area Chairs,
>
> As the discussion deadline approaches, we kindly request your assistance in reminding the reviewers to provide their post-rebuttal responses. We have made every effort to address their concerns thoroughly.
>
> Currently, Reviewer hAA1 has not yet responded to the authors. The reviewer mentioned, "Please refer to my questions above. I may have more questions during the discussion phase," but we have not received any further feedback. We wonder if our responses have adequately addressed hAA1's concerns. Should Reviewer hAA1 have any additional questions or concerns, we are fully committed to addressing them promptly and thoroughly.
>
> Besides, Reviewers GTSo and dQ8T have acknowledged that our responses have addressed their questions well and are willing to raise the scores (4->5 and 6->7) for the manuscript. Reviewer x7i3 points out some remaining concerns, and we actively provide detailed replies and are now awaiting the experimental results and feedback.
>
> Thank you for your assistance!
>
> Best regards,
>
> Authors of Submission 7269

---

### Decision · Program_Chairs · 2024-09-25

**Decision:**

Accept (poster)

**Comment:**

Very slightly over the threshold. Reviewers mostly praise the writing, presentation, experiments & study design, though have concerns about
-- Alignment between claimed contribution and experiments
-- Annotation and computational costs
-- Some issues of correctness (though somewhat addressed in rebuttal)
I'll also note that the reviewers don't read all that positively, in spite of marginally positive scores, and are on the short side, though the discussion is quite lengthy and adds more detail than what is in the reviews.